



# Vertical profiles of NO2, SO2, HONO, HCHO, CHOCHO, and aerosols derived from MAX-DOAS measurements at a rural site in the central-western North China Plain and their relation to emission sources and effects of regional transport

Yang Wang[1], Steffen Dörner[1], Sebastian Donner[1], Sebastian Böhnke[1], Isabelle De Smedt[2], Russell R. Dickerson[3], Zipeng Dong[4], Hao He[3], Zhanqing Li[3,4], Zhengqiang Li[5], Donghui Li[5], Dong Liu[6], Xinrong Ren[3,7], Nicolas Theys[7], Yuying Wang[4], Yang Wang[4], Zhenzhu Wang[6], Hua Xu[5], Jiwei Xu[6], Thomas Wagner[1]

1 Max Planck Institute for Chemistry, Mainz, Germany

2 Belgian Institute for Space Aeronomy (BIRA-IASB), Brussels, Belgium

3 Department of Atmospheric and Oceanic Science and ESSIC, University of Maryland, College Park, Maryland, USA

4 College of Global Change and Earth System Sciences, Beijing Normal University, Beijing, China

5 Institute of Remote Sensing and Digital Earth, Chinese Academy of Sciences

6 Key Laboratory of Atmospheric Composition and Optical Radiation, Anhui Institute of Optics and Fine Mechanics, Chinese Academy of Sciences, Hefei, China

7 Air Resources Laboratory, National Oceanic and Atmospheric Administration, College Park, Maryland, USA

*Correspondence to: Yang Wang (y.wang@mpic.de)*

## Abstract

A Multi Axis Differential Optical Absorption Spectroscopy (MAX-DOAS) instrument was deployed in May and June 2016 at a monitoring station (37.18 ° N, 114.36 ° E) in the suburban area of Xingtai (one of the most polluted cities in China) during the Atmosphere-Aerosol-Boundary Layer-Cloud ($A^2BC$) and Air chemistry Research In Asia (ARIAs) joint experiments to derive tropospheric vertical profiles of NO2, SO2, HONO, HCHO, CHOCHO and aerosols. Aerosol optical depths derived from MAX-DOAS were found to be consistent with collocated sun-photometer measurements. Also the derived near-surface aerosol extinction and HCHO mixing ratio agree well with coincident visibility meter and *in situ* HCHO measurements, with mean HCHO near-surface mixing ratios of ~3.5 ppb. Underestimates of MAX-DOAS results compared to *in situ* measurements of NO2 (~60%), SO2 (~20%) are found expectedly due to vertical and horizontal inhomogeneity of trace gases. Vertical profiles of aerosols and NO2, SO2 are reasonably consistent with those measured by a collocated Raman Lidar and aircraft spirals over the station. The deviations can be attributed to differences in sensitivity as a



function of altitude and substantial horizontal gradients of pollutants. Aerosols, HCHO, and CHOCHO profiles typically extended to higher altitudes (with 75% integrated column located below ~1.4km) than did $NO_2$, $SO_2$, and HONO (with 75% integrated column below ~0.5 km) under polluted condition. Lifted layers were systematically observed for all species, (except HONO), indicating accumulation, secondary formation, or long-range transport of the pollutants at higher altitudes.

Maximum values routinely occurred in the morning for $NO_2$, $SO_2$, and HONO, but around noon for aerosols, HCHO, and CHOCHO, mainly dominated by photochemistry, characteristic upslope/downslope circulation and PBL dynamics. Significant day-to-day variations are found for all species due to the effect of regional transport and changes in synoptic pattern analysed with HYSPLIT trajectories. Low pollution was often observed for air masses from the north-west (behind cold fronts), and high pollution from the southern areas such as industrialized Wuan. The contribution of regional transport

for the pollutants measured at the site during the observation period was estimated to be about 20% to 30% for trace gases, and about 50% for aerosols. In addition, agricultural burning events impacted the day-to-day variations of HCHO, CHOCHO and aerosols.

## 1 Introduction

The North China Plain (NCP) is one of the most populated, industrialized, and economically developed regions in

China. The NCP region is located in the northern part of eastern China with an area of about 3% of the total area of China and with about 20% of the Chinese population, covering major parts of the provinces Hebei, Henan, Shandong, the northern parts of Anhui and Jiangsu, and the megacities of Beijing and Tianjing. The NCP region is between the Bohai and Huanghai Seas to the east and the Taihang Mountains to the west. The Yan Mountains and the Dabie Mountains and Yangzi River delineate northern and southern boarders. With rapid economic growth and urbanisation, air pollution in the NCP region has

become severe. The NCP has suffers from the most frequent and severe haze events in China based on the reports from the Ministry of Environmental Protection (MEP, 2017). Previous studies characterized the composition of aerosol particles (e.g. Huang et al., 2014, Wang et al., 2012 and 2015b) and their gaseous precursors (e.g. Zhu et al., 2016, Hendrick et al., 2014) to better understand haze events (e.g. Fu et al., 2014). The role of regional transport (e.g. Ding et al., 2009) in haze events has been studied with chemical transport modelling (e.g. Wang L. et al., 2012 and 2015) and observations such as ground-

based stations, mobile platforms (e.g. Zhu et al., 2016), aircraft (e.g., Ding et al., 2009), and satellites (e.g. Tao et al., 2012). Previous studies demonstrated that secondary aerosols formed through photochemical reactions from trace gas precursors, e.g. nitrogen dioxide ($NO_2$), sulphur dioxide ($SO_2$), and volatile organic compounds (VOCs) contribute significantly to aerosols (e.g. Huang et al., 2014). Meteorology (e.g. transport patterns and mixing processes) is also critical for the formation of haze events (e.g. Miao et al., 2015, Wang L. et al., 2012, Li et al., 2007). Previous studies were usually based

on surface measurements of trace gases and and/or column densities derived from satellite observations. Observations of the vertical distribution of trace gases are also important to understand effects of chemical reactions, their sources and sinks, the influence of regional transport, and to validate results from chemical models and satellite observations. East-Aire and



associated campaigns generated a number of profiles from aircraft measurements and studied the vertical distribution to help understand the budgets of trace species and to aid in retrievals for remote sensing (Chaudhry et al., 2007; Dickerson et al., 2007; He et al., 2012; Krotkov et al., 2008; C Li et al., 2010; Z Q Li et al., 2007a; Z Q Li et al., 2007b). To date, profile measurements remain scarce.

The multi-axis differential optical absorption spectroscopy (MAX-DOAS) technique, invented about 15 years ago, allows one to derive vertical profiles of trace gases and aerosols in the troposphere from the observation of scattered sunlight at multiple elevation angles (Hönninger and Platt, 2002; Bobrowski et al., 2003; Van Roozendael et al., 2003; Hönninger et al., 2004; Wagner et al., 2004; Wittrock et al., 2004). The existing profile inversion approaches for MAX-DOAS can be sorted into two groups: algorithms based on the optimal estimation (OE) method (Rodgers, 2000; Frieß et al., 2006, 2011;
Wittrock, 2006; Irie et al., 2008, 2011; Clémer et al., 2010; Yilmaz, 2012; Hartl and Wenig, 2013; Wang Y. et al., 2013a, b) and the so-called parameterized approaches using look-up tables (Li et al., 2010, 2013; Vlemmix et al., 2010, 2011; Wagner et al., 2011; Irie et al., 2008, 2011). The MAX-DOAS technique is suitable for long-term observations of trace gases and aerosols with a relative high time resolution, several minutes, due to its simple instrument concept, low-cost, and automatic operation. Several networks of MAX-DOAS instruments have been built to record long-term measurements (e.g. Kanaya et
al., 2014). Such measurements and also data from short-term measurement campaigns have been used for environmental studies, as well as for the validation of satellite observations and model simulations (e.g. Irie et al., 2008; Roscoe et al., 2010; Ma et al., 2013; Kanaya et al., 2014; Vlemmix et al., 2015a; Wang T. et al., 2014; Wang Y. et al., 2017a, 2017b; Hendrick et al., 2014).

    Previous studies have reported MAX-DOAS measurements of $NO_2$, $SO_2$, nitrous acid (HONO), formaldehyde
(HCHO), and glyoxal (CHOCHO) in polluted regions in China (e.g. Wang T. et al., 2014; Hendrick et al., 2014; Wang Y. et al., 2017a; Li et al., 2013; Ma et al., 2013; Schreier et al., 2015). $NO_2$ and $SO_2$ can be converted to nitrate and sulfate, and $NO_2$ contributes to ozone formation. HONO is an important precursor of OH. Recent field measurements (e.g. Su et al., 2008, 2011 and Li et al., 2014, and references therein) suggest that the observed daytime HONO concentrations cannot be explained by the gas-phase reaction of NO with OH (Stuhl and Niki, 1972; Pagsberg et al., 1997); primary emissions of
HCHO could be important in industrial regions (Chen et al., 2014). HCHO and CHOCHO are mainly produced through the oxidation of VOCs, thus their high concentrations indicate photochemical activity. VOCs impact the formation of ozone and organic aerosols. CHOCHO and HCHO have different VOC precursors and different formation pathways (e.g., Vrekoussis et al., 2010 and Li et al., 2013).

    The "Atmosphere-Aerosol-Boundary-Cloud Interactions ($A^2BC$)" campaign took place in the heavily polluted
southern area of Hebei province from 25 April to 15 June 2016. The aim of the campaign was the investigation of the interaction of gas precursors, aerosols, and clouds from ground based and aircraft measurements. At the Xingtai measurement station in a rural area near Xingtai City, we operated a MAX-DOAS instrument developed by the Max-Planck-Institute for Chemistry (MPIC) to measure vertical profiles of aerosols, $NO_2$, $SO_2$, HONO, HCHO, and CHOCHO. The southern area of the Hebei province around the Xingtai station is in the central-west part of the NCP and contains several



cities ranked among the most polluted in China (based on reports from the Ministry of Environmental Protection), such as Xingtai, Shijiazhuang, Baoding, Tangshan, and Handan. Xingtai City with a population of about 7 million is frequently ranked as China's most polluted city (MEP, 2017). The Taihang Mountains at the west edge of the NCP are 30 km away from the Xingtai station. Previous studies demonstrated the effects of the Taihang Mountain on the accumulation and

dispersion of aerosols (e.g. Wei et al., 2010; Miao et al., 2015; Wang L. et al., 2015). MAX-DOAS measurements at the Xingtai station characterize the vertical distribution and temporal variation of aerosols and trace gases to better understand emission sources and effects of regional transport. The availability of other measurements of air pollutants during the campaign enhances the MAX-DOAS results. Spiral up-down aircraft measurements of trace gases and aerosols over the Xingtai station can be used to evaluate the vertical profiles retrieved from the MAX-DOAS measurements. MAX-DOAS

results can also be compared with surface concentrations derived from in situ measurements, aerosol optical depths (AODs) measured by a sun-photometer, and vertical profiles of aerosol extinction measured by a Raman Lidar. In parallel to A$^2$BC, the ARIAS campaign (Air chemistry Research In Asia) investigated trace gases, aerosols and cloud interactions over Hebei (Wang et al., 2018; Benish et al., 2018). These and satellite monitoring (Li et al., 2017) indicate a distinct downward trend in SO$_2$ over China recently.

This paper is structured as follows. Section 2 gives an overview on the topography and pollution conditions in the area around the measurement station and the MAX-DOAS measurements. Section 3 introduces other independent measurements and the trajectory simulations. Section 4 presents comparisons of MAX-DOAS results with independent measurements. Effects of regional and local transport of pollutants are discussed in section 5. The conclusions are presented in section 6.

**2 MAX-DOAS measurements**

**2.1 Overview of the measurement station**

MAX-DOAS measurements were performed during the A$^2$BC campaign at a station on the central-west edge of the NCP area (37.18 °N, 114.37 °E). Fig. 1a indicates the station in the southern area of the Hebei province surrounded by the provinces Shandong, Henan, and Shanxi. The Taihang Mountains are ~30 km west of the station. The terrain height

increases from ~ 10 m asl to ~1 km in the foothills of the Taihang Mountains. Winds from the mountains occurred frequently near the station during the measurement period. Information on wind speed and direction derived from a local weather station is shown in the Fig. S1 in the supplement. Surface winds reflect midday upslope (southeast) and nighttime downslope (northwest) circulation. Minimum wind speeds occur usually in a period between about of ~8:00 to 10:00 BT (GMT +8hr). Fig. 1b indicates the downtown area of Xingtai, with about 7 million inhabitants, located ~20 km southeast of the station. A

steel mill is located south-west of the Xingtai downtown area. A large industrial area with several steel and coal coking facilities is located near Wuan City, ~50 km south and southeast of the measurement station. A satellite image of the



industrial area derived from the 'Google Maps' service (https://www.google.de/maps) is shown in Fig. 1c, and a zoomed image of one of the factories  (Fig. 1d) shows many stacks.

Averaged maps of $NO_2$, $SO_2$, and HCHO derived from satellite observations of the Ozone Monitoring instrument (OMI) (Levelt et al., 2006a and b) for May and June during the period 2012 to 2016 for the same area as shown in Fig. 1a

are shown in Fig. 2a, b, and c, respectively. In Fig. 2d a map of the averaged aerosol optical depths (AODs) at 550 nm derived from the Moderate Resolution Imaging Spectroradiometer (MODIS) (Kaufman et al., 2002) for the same period is shown (provided by NASA on http://ladsweb.nascom.nasa.gov/data/search.html). A strong decrease of all four pollutants along the Taihang Mountains at a line from northwest to southeast is obvious. The measurement station is located in the polluted region, but close to its edge. The patterns of HCHO and AOD are more homogenous than $NO_2$ and $SO_2$. Large

values of $NO_2$, $SO_2$ and AOD can especially be found in the heavily industrial Wuan area. High amounts of $NO_2$ and AOD, but not $SO_2$, can be seen in the north of the station at a distance of about 100 km around Shijiazhuang, the capital of the Hebei province with about 11 million habitants.

## 2.2 Operation and processing of MAX-DOAS measurements

### 2.2.1 Measurement operation

A "Tube MAX-DOAS" instrument (Donner, 2016) developed by MPIC, Mainz, Germany, was operated at the measurement station during the period from 8 May to 10 June 2016. More details about the instrument can be found in Donner et al. (2016) and Wang et al. (2018). Spectra of scattered sunlight were routinely recorded by the MAX-DOAS instrument at 11 elevation angles (1 °, 2 °, 3 °, 4 °, 6 °, 8 °, 10 °,15 °, 20 °,30 °, 90 °) in the wavelength range of 300 to 466 nm with a spectral resolution of about 0.6 nm.  The exposure time of each individual spectrum was ~1 minute. Electric offset and

dark current are corrected using background measurements taken at night.

### 2.2.2 DOAS retrievals of slant column densities

Differential slant column densities (dSCDs) (the integrated trace gas number density along the effective light path) of $NO_2$, $SO_2$, HCHO, HONO, and CHOCHO, and the oxygen dimer ($O_4$), are retrieved from the recorded spectra using the DOAS technique (Platt and Stutz, 2008). The analysis is implemented using the QDOAS software (http://uv-

vis.aeronomie.be/software/QDOAS/) (Danckaert et al., 2017). The DOAS retrievals are configured based on previous studies, e.g. Wang et al. (2017a, c), and for the MAD-CAT campaign (http://joseba.mpch-mainz.mpg.de/mad_analysis.htm). The settings are listed in Table 1.  A spectrum measured in the zenith direction closest in time to the off-zenith measurements is used as a Fraunhofer reference spectrum. Typical examples of DOAS fits of the six species are given in Fig. 3. The root mean square (RMS) of the optical depth of the fit residuals are typically around $6\times10^{-4}$ for $O_4$, $NO_2$, HCHO and HONO,

$2\times10^{-4}$ for CHOCHO, and $1\times10^{-3}$ for $SO_2$. The detection limits for the dSCDs are about $2\times10^{15}$, $6\times10^{15}$, $3\times10^{15}$, $1\times10^{15}$,



$4 \times 10^{14}$ molecules cm$^{-2}$ for NO$_2$, SO$_2$, HCHO, HONO, and CHOCHO, respectively, and $6 \times 10^{41}$ molecules$^2$ cm$^{-5}$ for O$_4$ under typical measurement conditions.

### 2.2.3 Retrieval of vertical profiles, near-surface values, and vertical column densities

Tropospheric vertical profiles of aerosol extinction and volume mixing ratios (VMRs) of NO$_2$, SO$_2$, HONO, HCHO, and CHOCHO are retrieved from the elevation-dependent dSCDs by using the PriAM profile inversion algorithm (Wang et al., 2013a, b, 2017a). From the derived profiles the vertical column densities (VCD) of the trace gases are derived by vertical integration. The PriAM algorithm is based on the optimal estimation method (Rodgers, 2000) using the radiative transfer model (RTM) of SCIATRAN version 2.2 (Rozanov et al., 2005). The inversion consists of two steps: first, vertical profiles of aerosol extinctions are retrieved from the measured O$_4$ dSCDs; second, the retrieved aerosol profiles are used for the inversion of the trace gas profiles. For the radiative transfer simulations the surface height and surface albedo are set as 150 m a.s.l. and 0.05, respectively. A fixed single scattering albedo (SSA) of 0.95 and an aerosol phase function parameterised according to Henyey and Greenstein (1941) with an asymmetry parameter of 0.72 are chosen according to averaged inversion results from the sun-photometer also operated at the measurement station. While Wang et al., (2018) reported an averaged SSA of 0.85 at 550 nm based on aircraft observations, the differences between aircraft and the sun-photometer measurements is not currently understood. The systematic bias of SSA typically contributes to an uncertainty of about 5% to retrieved aerosols and trace gases from MAX-DOAS measurements.

Retrieved aerosol extinction profiles at 360 nm are converted to those at 313 nm, 339 nm, and 354 nm for SO$_2$, HCHO, and HONO, respectively, using an Ångström exponent of 1 as derived from measurements of a collocated sun-photometer (see section 3.1.1). Because of a variability of Ångström exponent and uncertainties of sun-photometer measurements (see discussion in section 3.1.1), the Ångström exponent could typically contribute uncertainties of up to 20% to retrievals of trace gas profiles. From the MAX-DOAS inversion, profiles below 3 km are derived. Temperature and pressure profiles are derived from the MPIC climatological data base. Different a priori profiles are used for the individual species according to previous studies (Wang et al, 2017a and Hendrick et al, 2014) and/or sensitivity tests using different a-priori profiles. The a-priori profiles are shown as the grey curves in Fig. 4d. The diagonal elements of the a-priori covariances (S$_a$) at different altitudes are set as the square of 100% of the a-priori values at the surface in order to balance the flexibility and stability of the profile inversion.

One indicator for the confidence of the profile inversion is the consistency of the measured and modelled dSCDs (dSCDs simulated by the RTM SCIATRAN for the retrieved profiles). For a systematic analysis, we screened the suspicious profile results with larger differences of measured and modelled dSCDs than the thresholds listed in Table 2. After the filtering, the scatter plots, correlation coefficients (R), and slopes derived from linear regressions of the measured against the modelled dSCDs for the different species during the entire measurement period are given in Fig. S2 in the supplement. The correlation coefficients R are higher than 0.95 and the slopes deviate from unity by less than 2% for all species.





11 May 2016, a typical day with high pollution was selected to show MAX-DOAS results for polluted conditions. Time series of retrieved profiles from the MAX-DOAS measurements on 11 May 2016 are shown in Fig. 4b, and selected profiles around noon are plotted in Fig. 4d. Note that profiles shown in Fig. 4b are not screened based on the differences of modelled and measured dSCDs. However, the black dots in the top of each panel of Fig. 4b indicate the confident results that remain after filtering). In Fig. 4b a large variability of profile shapes and absolute values can be seen, especially at altitudes below 1km. The sensitivity of the profile retrievals of trace gases and aerosols systematically decrease with altitude. The typical averaging kernels of the profile inversions shown (Fig. 4c) indicate the effect.

### 2.2.4 Cloud classifications from MAX-DOAS measurements

Since clouds can strongly impact the MAX-DOAS results, different sky conditions are identified from the MAX-DOAS observations using the cloud classification scheme described in Wang et al. (2015) and Wagner et al. (2014). The scheme assigns individual MAX-DOAS measurements to six sky condition categories: 'cloud free and low aerosol load, 'cloud free and high aerosol load, 'cloud holes', 'broken clouds', 'continuous clouds', and 'optically thick clouds'. The derived sky conditions are indicated by the colour dots in the Fig. 4a. Based on the study by Wang et al. (2017a), cloud contaminated results of aerosol profiles and AODs are not included in the further analysis; but for all other MAX-DOAS results (including all trace gas results and near-surface aerosol extinctions), only the data under the 'optically thick clouds' condition are skipped.

### 2.2.5 MAX-DOAS results during the entire measurement period

Figure 5 presents an overview of the near-surface values and column densities derived from the MAX-DOAS measurements during the campaign from 8 May to 10 June. To provide some information about the diurnal variation, daily averages for three-time intervals of 6-10h (morning), 10-14h (noon), and 14-18h (afternoon) are shown. The corresponding full time series of the MAX-DOAS results for individual days are shown in Fig. S3 in the supplement. To remove measurements of reduced quality filters for SZA, relative intensity offset, RMS of the residuals of the DOAS fits, differences of modelled and measured dSCDs, and sky conditions are applied to the results. The details of the filtering process and thresholds for different species are shown in Table 2.

Average near-surface aerosol extinctions (0.43 km$^{-1}$), and VMRs of $NO_2$ (7.8 ppb), $SO_2$ (17.1ppb), HONO (0.22 ppb), HCHO (3.33 ppb), and CHOCHO (0.08 ppb), are shown in Fig. 5 as are AOD (0.65) and VCDs of $NO_2$, (15.6 $\times 10^{15}$ molecules cm$^{-2}$) $SO_2$ (31.7 $\times 10^{15}$ molecules cm$^{-2}$), HONO (0.21 $\times 10^{15}$ molecules cm$^{-2}$), HCHO (13.7 $\times 10^{15}$ molecules cm$^{-2}$), and CHOCHO (0.32 $\times 10^{15}$ molecules cm$^{-2}$). Rather large day-to-day fluctuations are found for $NO_2$, $SO_2$, and HONO, especially in the morning, while the variations for aerosols, HCHO, and CHOCHO are smaller. This finding can probably be attributed to transport and the regional nature of secondary pollutants and will be further discussed in section 5. Maximum values for $NO_2$, $SO_2$, and HONO frequently occurred in the morning, but around noon for aerosols, HCHO, and CHOCHO. Again, this finding can probably be attributed to the different sources and deposition pathways of the different species. $NO_2$





and $SO_2$ are removed through reactions with the OH radical, which is more abundant during daytime than during nighttime. HONO is rapidly photolysed after sunrise. Therefore maximum concentrations of $NO_2$, $SO_2$, and HONO can be expected in the morning when depth of planetary boundary layer (PBL) is also low and emissions could be also high. While CHOCHO and HCHO are also removed via reaction with OH, they are also produced by the reaction of OH with different organic

compounds (HCHO can also be directly emitted). Because of the high correlation of HCHO and CHOCHO (with R of about 0.94), secondary formation of HCHO is probably the dominant source during the campaign. Typically HCHO and CHOCHO peak around noon indicating that the production rate is higher than the loss rate around noon. For particles, Zhang et al. (2018) also showed that secondary aerosols were dominant in the measurement area based on measurements of an Aerosol Chemical Speciation Monitor and a Scanning Mobility Particle Sizer located in the measurement station. Since also the

secondary formation of aerosols depends on OH radicals the maximum aerosol formation can be expected around noon. However, besides the effects of photochemistry, variations of anthropogenic emission rates, effects of local and regional transport, and dispersion can also impact the diurnal variations. These effects will be discussed in more detail in section 5.2.

      Consistent day-to-day variations can be found between $NO_2$, $SO_2$, and HONO, especially in the morning. We calculate the correlation coefficients (R) between the species because good correlations could imply similar sources and

sinks of the species, R in the morning (in the afternoon) between $NO_2$ and $SO_2$, $NO_2$ and HONO, and $SO_2$ and HONO are about 0.95 (0.7), 0.73 (0.25), and 0.73 (0.15), respectively. Those between HCHO and CHOCHO are about 0.87, 0.89, and 0.94 in the morning, around noon, and in the afternoon, respectively. Moderate correlations of the day-to-day variations can be found between all the species and aerosols in the morning with R of 0.6 to 0.7 probably indicating similar sources. In the afternoon, the correlation between aerosols and with HCHO is up to 0.75, but those with $NO_2$, $SO_2$ are below 0.5, again

reflecting the difference between primary and secondary pollutants. A recent study (Zhang et al., 2018) demonstrated that 78% of $PM_1$ aerosol particles are secondary aerosols based on the collocated measurements of aerosol composition. They also demonstrated that organic aerosols contribute to about 40% of total, consistent with the high correlation between aerosols and HCHO. Sources of the species will be further discussed in section 5.

      The correlation between the near-surface and column values is also investigated for the different species. Higher

correlations (R) are found for HCHO (0.9) and HONO (0.93) than for aerosols (0.76), $NO_2$ (0.83), and $SO_2$ (0.78). The moderate correlations of aerosols, $NO_2$ and $SO_2$ could be attributed to the frequent occurrence of lifted layers (see Fig. 12a, b and c) probably related to transport of pollutants. Remaining cloud effects in the profile inversion after cloud screenings applied could also partly contribute to the lifted aerosol layers retrieved from MAX-DOAS measurements. The characteristics of the vertical profiles of all species under different situations with different importance of pollution transport

will be discussed in section 5.

      Ratios of HONO and $NO_2$ have been often used to characterize possible sources of HONO (e.g. Sörgel et al., 2011a; Wojtal et al., 2011; Li et al., 2012). The day to day variations of the HONO/$NO_2$ ratios (VCDs and VMRs) in the morning, around noon, and in the afternoon are shown in Fig. 5. The ratios are between 1.9% to 2.9% on average, with higher ratios in the morning, and slightly larger than those (0.9% to 2.4%) found in spring and summer in Xianghe (a suburban area of



Beijing) between 2008 to 2013 (Hendrick et al., 2014). The frequent occurrence of the peak ratio HONO/NO$_2$ in the morning is consistent with the observation in Xianghe (Hendrick et al., 2014) and could be explained by the faster removal of HONO than of NO$_2$. Note that Yu et al. (2009) reported HONO/NO$_2$ ratios of up to 30% during night time based on long-path DOAS measurements in Kathmandu, Nepal. Another interesting finding of our study is that on several days (e.g. on 15 May

and 6 June), the noon HONO/NO$_2$ ratios are even up to 10%. This finding is coincident with low NO$_2$ values, indicating that the known typical daytime HONO source (gas phase reaction of NO and OH) cannot explain the observed relatively high HONO values.

Ratios of CHOCHO and HCHO have been used in a number of previous studies to identify sources of VOCs since CHOCHO and HCHO have different precursors or different formation pathways (e.g., Vrekoussis et al., 2010 and Li et al.,

2013). The day to day variations of the ratios CHOCHO/HCHO (VCDs and VMRs) in the morning, around noon, and in the afternoon are shown in Fig. 5. They are between 2% to 2.5% on average. Similar ratios have been observed at rural sites, e.g. 1.7% in Nashville, USA (Lee et al., 1998), 3.6% in Cabauw, The Netherlands (Irie et al., 2011) and at urban sites, e.g. 3.6% in Mexico City, Mexico (Lei et al., 2009). However, considerably higher ratios of up to 10% were also reported in previous studies. For instance, averaged ratios of 6% to 8% were derived from the MAX-DOAS measurements in July 2006 in the

suburban area of Guangzhou city in Southern China (Li et al., 2013). The lower ratios derived from our measurements could be (at least partly) be related to anthropogenic primary emissions of HCHO. Measurements shown in Benish et al. (2018) also indicate isoprene, a dominant nature source of secondary HCHO, is not high in the measurement area.

## 3 Independent data

The MAX-DOAS results are compared to several independent ground-based measurements at the measurement

station as well as aircraft measurements over the station. The independent measurements are introduced in section 3.1. For the interpretation of the MAX-DOAS results with respect to transport, we performed backward trajectory simulations and also used the meteorology data from a local weather station. Both data sets are introduced in section 3.2.

### 3.1 Independent measurements for comparisons with MAXDOAS results

### 3.1.1 In-situ measurements of NO$_x$, SO$_2$, HCHO, Sun photometer and visibility meter

An in-situ gas analyser system manufactured ECOTECH measured VMRs of CO, NO, NO$_x$, SO$_2$, and O$_3$ with a time resolution of ~3 s during the whole measurement period. NO$_2$ VMRs are derived by subtracting NO VMRs from NO$_x$ VMRs.    For comparisons with MAX-DOAS results, the in-situ measurements are averaged over the individual time intervals of the MAX-DOAS measurements.

Surface HCHO VMRs were monitored during the period from 18 to 23 May, 2016, using the Formaldehyde

Analyzer (AERO LASER, Germany, Model 4021) based on fluorometric Hantzsch reactions [*Gilpin et al.*, 1997; *Rappenglü*




*ck et al.*, 2010] with a time resolution of about 1 min. For the comparison with the MAX-DOAS HCHO results, the in-situ measurements are averaged over the individual time intervals of the MAX-DOAS measurements.

A sun photometer operated by the institute of Remote Sensing and Digital Earth, Chinese Academy of Sciences, measures the AODs at eight wavelengths between 340 nm and 1634 nm. The AODs at 340 nm and 380 nm are averaged for the comparison with the AODs retrieved from MAX-DOAS measurements at ~360 nm. The Ångström parameter, single scattering albedo, and asymmetry factor are also retrieved from the sun-photometer measurements. The single scattering albedo, and asymmetry factor are used for the inversion of the aerosol profiles from MAX-DOAS measurements, see section 2.2.3.

A forward-scattering visibility meter (550 nm) was also operated at the measurement station during the entire measurement period. Aerosol extinction at 360 nm is derived from the visibility at 550 nm using a typical (Sun et al., 2018) parameter of 1 derived from the sun photometer measurements. The conversion could contribute a typical uncertainty of up to 20% due to variability and uncertainties of the Ångström exponent. Wang et al. (2018) reported a much higher value for the Ångström exponent: from 0.49 to 2.53 (median 1.53) for the same campaign. These values are based on observations made from the aircraft equipped with an aerosol inlet with a reported 50% cutoff at 5 μm. The decreased sensitivity for large aerosol particles together with unobserved aerosol below the lowest altitude of the aircraft spirals (~300m agl) could contribute to the difference between the aircraft and ground-based sun photometer measurements. Sun photometers measure AOD directly, and the Ångström exponent is derived from AODs at different wavelengths.

### 3.1.2 Lidar

A three-wavelength Raman Polarization Lidar system (Tao et al., 2012 and Liu et al., 2013) developed by the Key Laboratory of Atmospheric Composition and Optical Radiation, Hefei Institute of Physical Science, CAS, was operated on several days during the campaign. Profiles above 500m of the aerosol extinction at 355 nm are retrieved from the Lidar measurements. There are two mainly cloud-free days (16 and 17 May), on which the Lidar measurements overlapped with the MAX-DOAS measurements. Therefore comparisons between MAX-DOAS and Lidar are done for these days (section 4.2).

### 3.1.3 Aircraft measurements

On several days during the measurement period, a Y-12 airplane (twin engine multi-purpose transport aircraft, Harbin Aircraft Manufacturing Corporation) from the Weather Modification Office of the Hebei Meteorological Bureau flew spirals down to about 200 m above the measurement station and other sites to obtain atmospheric profiles of aerosol optical properties and trace gases concentrations (*Wang et al.*, 2018 and Benish et al., 2018). The diameters of the spiraling circles were about 10 km. The aircraft measurements overlapped with the MAX-DOAS measurements on 8 and 21 May 2016. On the Y-12 airplane, $SO_2$ was monitored by a commercially available trace level pulsed fluorescence analyzer (TEI Model 43C) (Luke, 1997). $NO_2$ was measured using a modified commercially available cavity ring-down spectroscopy (CRDS) detector



(Brent et al., 2013; Castellanos et al., 2009). Aerosol scatterings were measured at 550 nm using an integrating Nephelometer (Trust Science Innovation, TSI Model 3563) and aerosol absorptions were measured at 565nm and converted to that at 550 nm using a Particle Soot Absorption Photometer (PSAP) (e.g. Anderson et al., 1996; Taubman et al., 2006). All instruments were routinely calibrated during the measurements (Brent et al., 2013; He, 2012; He et al., 2014; Taubman et al.,

2006). For the comparison with the MAX-DOAS profiles, the aerosol extinction at 550 nm (the sum of aerosol scattering and absorption) is converted to that at 360 nm using averaged Ångström parameters of 0.8 on 8 May and 1.5 on 21 May derived from the sun photometer measurements.

## 3.2 Meteorological data and air mass trajectories

To interpret the measurement results, we derived meteorological parameters including ambient temperature (T),
relative humidity (RH), wind speed (WS), wind direction (WD), and precipitation from the weather station at the measurement station of the MAX-DOAS instrument. The weather station was conducted by the Xingtai Meteorological Administration. For the discussion of the effect of regional transport of pollutants in section 5, we performed 12 h backward trajectory simulations starting at each hour throughout the day and ending at 100 m, 300 m and 1km above the measurement station using the HYSPLIT model (Stein et al., 2015) developed by Air Resources Laboratory, NOAA
(https://ready.arl.noaa.gov/HYSPLIT.php). The meteorological data used for the backward trajectory calculation was from the NCEP Global Data Assimilation System (http://www.emc.ncep.noaa.gov/gmb/gdas/) with a spatial resolution of 0.5 degrees and a time resolution of 6 h.

## 4 Comparison of MAX-DOAS results with independent measurements

### 4.1 Comparison with a sun-photometer, visibility-meter, and in-situ measurements

Comparisons of MAX-DOAS aerosol results with co-located independent measurements are done under three different sky condition categories, including "clear sky with low aerosol load", "clear sky with high aerosol load", "cloudy sky" (optically thick clouds are excluded). The sky conditions are characterised based on MAX-DOAS measurements (see section 2.2.4). The comparison results are shown in Fig. 6.

AODs retrieved by MAX-DOAS measurements are compared with those retrieved from the sun-photometer
measurements in Fig. 6a. Because for MAX-DOAS measurements AOD results are skipped for cloudy conditions (and also sun photometer measurements are only available for clear sky), no comparison results for cloudy situations are shown in Fig. 6a. For clear sky observations, better agreement is found for the category "low aerosol" (R of 0.84 and slope of 1.13) than for "high aerosol" (R of 0.71 and slope of 0.65). Probably, the a-priori constraint in the MAX-DOAS profile inversion is the main reason for the underestimation of the AOD under "high aerosol" conditions. The comparison of the near-surface
aerosol extinction between the MAX-DOAS measurements and the visibility-meter are shown in Fig 6b. While very good agreement is found for the category "clear sky with low aerosols" (R of 0.81 and slope of 1.02), much worse correlation is



found for "clear sky with high aerosols" (R of 0.32 and slope of 0.14). This finding probably indicates the effect of elevated aerosol layers (see also Fig. 4 d), which affect the near surface extinction values derived from MAX-DOAS measurements due to the low vertical resolution. Even worse agreement is found for the category "cloudy sky" (R of 0.22 and slope of 0.11), most probably because of the effects of clouds on the MAX-DOAS aerosol retrieval. Note that clouds are not included in the RTM which is used for profile retrievals of aerosols and trace gases. The limited vertical resolution of the MAX-DOAS results for aerosol extinction also probably plays a role for the comparison of MAX-DOAS near-surface trace gas results with in-situ measurements in the following paragraph.

The near-surface $NO_2$ and $SO_2$ VMRs retrieved from the MAX-DOAS measurements are compared to the in-situ measurements (see Sect 3.1.1) in Fig. 6c and d, respectively. Rather high correlation coefficients are found for both $NO_2$ (R of 0.9) and $SO_2$ (R of 0.95) for the category "clear sky with low aerosol". Larger scatter is found for the other two sky conditions partly due to cloud effects on the MAX-DOAS profile inversion. However, it is in general also found that the MAX-DOAS results are systematically lower than the in-situ results for $NO_2$ and $SO_2$ (slopes below unity). This finding is probably related to vertical and horizontal inhomogeneity of the species, different air measured by the two techniques, some artefacts of spectral analysis and profile inversion of MAX-DOAS measurements. In addition interference of $NO_y$ with $NO_x$ in the in-situ measurements could cause overestimation of $NO_2$. This might explain the smaller slope of $NO_2$ than of $SO_2$ in Fig. 6. The correlation plot of the near-surface HCHO VMRs retrieved from the MAX-DOAS measurements versus those from the in-situ HCHO analyser is shown in Fig. 6e. Note that only one week of in situ data is available for this comparison. And the category "clear sky with low aerosols" did not occur in this week. The averaging values are comparable between the two measurements. However, the slopes of ~0.3 strongly deviate from unity similar with other species.

## 4.2 Comparisons with Lidar measurements

In this section profiles of aerosol extinction retrieved from the MAX-DOAS measurements are compared with those derived from the Lidar measurements on two mostly cloud-free days. Aerosol loads on 16 May are lower than those on 17 May. The comparison of the time series of both selected days is shown in Fig. 7, also including the AODs derived from the MAX-DOAS and the sun-photometer measurements. Note that no aerosol profiles below 500 m are derived from the Lidar measurements due to missing overlap between the outgoing beam and the FOV of the telescope. In general reasonable agreement of the aerosol profiles from both techniques is found above 500 m. The remaining differences can probably be explained by the fact that different air masses are observed by both techniques and horizontal inhomogeneity of aerosols and cloud cover could appear, namely different clouds and aerosols were observed by both instruments. Note that the MAX-DOAS telescope was pointed towards the North, while the Lidar measured the atmosphere directly above the station. The sun-photometer measured the air masses in the direction of the sun.



### 4.3 Comparisons with aircraft measurements

The MAX-DOAS results of vertical profiles of aerosol extinction and $NO_2$ and $SO_2$ VMRs are compared to the aircraft measurements over the station on 8 and 21 May 2016. The individual and averaged values of the aircraft measurements as well as the averaged, minimum, and maximum profiles derived from the MAX-DOAS measurements during the overpass times of the aircraft are shown in Fig. 8 (also including the surface values measured by co-located independent measurements). To account for the smoothing effect of the MAX-DOAS profile inversion, the averaging kernels of the MAX-DOAS profiles (see Fig. 4c) are applied to the averaged aircraft profiles (referred to as "smoothed profiles" in the following) and the combined profiles derived from the averaged aircraft profile and surface data (referred to as "smoothed combined profiles" in the following). In general the "smoothed profiles" of all the species agree better with the MAX-DOAS results than the original aircraft profiles. And the values at the height of 300 m in the "smoothed combined profiles" are often higher than those in the "smoothed profiles" due to contributions of surface values to high altitudes by the smoothing of the averaging kernels. The effect of the smoothing is especially strong on 21 May. On that day, the box-shape profiles of aerosols and $SO_2$ below 2 km are 'reshaped' to exponentially decreasing profiles with maxima at about 500 m altitude. Besides errors of the MAX-DOAS profile inversion, the deviations between the MAX-DOAS and aircraft measurements can probably also be attributed to inhomogeneous horizontal distributions of pollutants and their temporal variation during a period of aircraft measurements. The differences between the maximum and minimum profiles from the MAX-DOAS measurements in Fig. 8 indicate the range; the scatter of the original aircraft measurements indicates considerable horizontal gradients within the ~10 km diameter of the spirals.

## 5 Regional and local transport of pollutants

In this section, effects of regional and local transport of pollutants in the measurement area are discussed based on a case study in section 5.1 and a systematic analysis in section 5.2. Possible cleaning effects of pollutants caused by precipitation are also discussed in in Sect 5.1.

### 5.1 A typical case period

Photos taken by a camera along the line of sight of the MAX-DOAS instrument around noon on the days from 11 to 16 May, 2016 are shown in Fig. 9, indicating different pollution conditions. What happened on these days to bring either blue skies or low visibility (high pollution)? The question will be answered based on regional and local meteorological data and the temporal evolution of the pollutants. The regional transports related to weather system of cold and warm front from the northwest and south could be the dominant driver for the air quality in the region.

All available data on pollutants and meteorological parameters during the period from 11 to 16 May 2016 are shown in Fig. 10. The corresponding plots for the other days of the campaign are shown in Fig. S3 in the supplement. 12 h





backward trajectories ending at 100 m and 1 km over the measurement site in intervals of 1 h are also shown in Fig. 10. The trajectories indicate the regions from which regional transport of air mass will originate at the measurement site. Also surface wind directions and wind speeds are provided in Fig. 10, indicating the origin of local (short-range) transport of air. The mountain-plain topography causes a daily upslope (southeast winds) downslope (northeast winds). Hourly accumulated

precipitation rates, ambient humidity and temperatures are also shown in Fig. 10. Note that trajectories, local winds, precipitation, humidity and temperatures are shown for the full 24 hours of each day, while visibility, AOD at 360 nm, surface trace gas VMRs and MAX-DOAS results are only shown for daytime (6:00 to 18:00 BT). The results derived from the MAX-DOAS measurements in Fig. 10 include cloud classification results and vertical profiles of aerosols and trace gases (including cloud contaminated results). Note that the profiles shown in Fig. 10 are not screened based on the differences of

modelled and measured dSCDs (see Table 2) and clouds, but the black dots in the top of each panel of profiles indicate the confident results which have passed the filters described in Table 2. In addition, daily total numbers of fire points derived from the MODIS satellite observations in the NCP area are shown in Fig. 10, in order to show the potential influence of biomass burning.

The results shown in Fig. 10 indicate that aerosols and trace gases steeply decrease from 11 to 12 May. Figure 9

indicates that also the visibility significantly increased from 11 to 12 May. The backward trajectories in Fig. 10 indicate that the origin of air masses arriving at the measurement site changes from south and south-west to north-west during the night from 11 to 12 May. As shown in Fig. 1 and 2, the area around Wuan about 50 km southwest of the measurement site is significantly polluted due to emissions from many iron and coal coking factories. The high amounts of pollutants observed in the morning on 11 May can be attributed to night-time transport of pollutants from the Wuan area based on the dominant

south-westerly trajectories before sunrise. In contrast the areas in the other directions, especially in the north-west, are relatively clean. Therefore the cleaning event can be attributed to regional transport of clean air from north-west.

The 14th was a day of steady, stratiform rain associated with a warm front followed by cold front passage from the northwest. Frontal passage on the 14th with southeast (SE) winds ahead of the front with northwest (NW) winds and high pressure behind it. The dominant NW and SE trajectories control the measurement area on the day before and after the

weather event on 14 May. Therefore a slightly higher pollution level was observed on 13 May than that on 12 May due to the transports of pollutants from south-east. A clean sky appeared on 15 May due to the transports of clean air mass from north-west. In addition, it should be noted that the heavy rain happened on 14 May could remove the $SO_2$, which is highly soluble in water, but not $NO_2$, which is less soluble. Considering wet deposition could have considerably impacted the pollution conditions. Therefore rainy days (14 May and 4, 5 June) are not considered in the discussion and in the following analysis

about the effects of regional transport.

On 16 May transport of pollutants from the south-west direction can be seen after 14:00 BT. However the concentrations of $NO_2$ and $SO_2$ during the transport event on 16 May are much lower than those in the morning of 11 May. The difference might be attributed to the shorter life time of $NO_2$ and $SO_2$ during daytime than nighttime. Therefore we can expect that the gas pollutants, e.g. $NO_x$, $SO_2$ and HONO, can be transported to a farther distance during nighttime than



daytime. Thus nighttime regional transport of pollutants from the Wuan area could significantly pollute the entire measurement area.

In summary we conclude that high pollution levels at the measurement site typically occur if air is transported from south. Low pollution is associated with other wind directions, especially if air is transported from the northwest briskly behind a cold front. In addition, Wang et al. (2018) demonstrated that particle formations from gas precursors and growths significantly contribute to aerosols and impact the visibility during the campaign. This effect is combined with transports of air mass because activities of particle formations depend on amounts of gas pollutants.

## 5.2 Systematic analysis of the origin of pollution during the campaign

### 5.2.1 Source areas of pollutants contributing via regional transport

In order to reveal the effects of transports from different areas on the variations of pollutants observed by MAX-DOAS, we applied a novel procedure based on the backward trajectories. In the procedure, first a grid map of the region around the measurement site within the latitude and longitude ranges of 4 °(~360km) is created with a spatial resolution of 0.1 °(~10 km) . Then we assign the observed column densities of pollutants at the measurement site to individual map pixels, from which are connected to the measurement site by backward trajectories. This procedure is performed for individual MAX-DOAS measurements throughout the entire campaign, and all values in the individual pixels are averaged to generate averaged pollution maps. In the reproduced maps, areas with high values of pollution imply the pollution emissions in the areas could be the dominant source of the high pollution observed at the measurement site via transports. The reproduced maps can be compared with satellite images. The consistencies between the reproduced maps and satellite maps imply transports could be the dominant source of pollutants and the dominant cause of the day-to-day variations of pollutants observed at the measurement site. The inconsistencies imply local emissions and local chemical reactions dominate the variability of pollutants. In order to consider pollution transport at different altitudes, trajectories ending at 100m, 300m, 500m, 1km, and 2km above the measurement site are used to create the maps. It needs to be note that similar approaches have been developed and applied to derive regional and global emissions of long-lived atmospheric trace gases and particles (e.g., halocarbons, hydrofluorocarbons, carbon monoxide, and black carbon) from in-situ measurements, e.g., Stohl et al., 2009, Brunner et al., 2012, and Xu et al., 2013. Emissions of the short-life trace gases are hard to determined using the approach due to variability of their life time. Therefore the approach is adapted in the study to only qualitatively analyse effects of pollution transports. The lifetime is only implicitly considered in the approach by using different backward time of trajectories for generating maps.

For $NO_2$ and $SO_2$, high values are found in the southwest (near Wuan city), and low values in the northwest of the measurement site. For HCHO and AOD, significantly higher values can be seen in the south and east compared to the northwest of the measurement site. The patterns in the created maps are generally consistent with the satellite maps indicating that air parcels transported from clean or polluted areas considerably contribute to the pollutant levels near the



measurement site. Interestingly, the patterns in the reproduced maps change by using different times of the backward trajectories. Correlation coefficients of the values in the reproduced maps against the satellite maps are also shown in Fig. 11 for the maps created for different times of the backward trajectories. Better agreement between the reproduce maps and the satellite maps is found for shorter backward times (1h) for $NO_2$, $SO_2$, AOD and longer backward times (8 h) for HCHO. This finding is probably related to the typically shorter life times of $NO_2$ and $SO_2$ than some VOCs, a source of secondary HCHO. It also indicates that the transport from closer sources is probably more important for NO2 and SO2 than the transport from a farther distance, vice versa for HCHO. For aerosols the dependence is much weaker than for the trace gases indicating comparable contributions of short-range and long-range transport to aerosols. In addition, in order to test effects of transport at different altitudes. The resulting maps for different altitudes, Fig. S4 in the supplement, indicate pollutants at different altitudes are mainly from the same source directions, but those at higher altitudes could be transported from farer areas.

### 5.2.2 Characteristics of the pollutants under different dominant trajectories

In order to quantify the differences of pollutants under different dominant transport conditions, we sort the measurement days of the whole campaign into three groups based on synoptic situation and the dominant directions of the nighttime trajectories, including southerly, north-westerly, and easterly trajectories. The sorting is also related to synoptic situations. The southerly and north-westerly trajectories are related to the warm sectors ahead of the front and cold sectors behind the front, respectively. The easterly trajectories are for the site controlled by a maritime tropical air mass. Considering that the life times of the observed trace gases are typically longer during night time than day time (because of lower OH radical concentrations), the measurement data are sorted mainly based on the nighttime trajectories. Nine days (09, 11, 18, 19, 23, 30, 31 May and 7, 9 June), eight days (12, 13, 15, 16, 17, 24, 25, 26 May), and eight days (20, 21, 22, 27 May and 1, 2, 6, 8 June) fall in these three categories with dominant southerly, north-westerly, and easterly trajectories, respectively. Figure S5 in the supplement presents contributions (percentages) of air mass from different locations in the area ($\pm 2\,°$Lat. and $\pm 2\,°$Lon. about $180 \times 180$ km$^2$) around the measurement station based on daytime and nighttime trajectories for the three groups of days. Figure S5 indicates that most of the trajectories come from the chosen dominant trajectory directions, but a few of them also come from other directions because of the changes of the wind fields during the day. In general, the dominant directions of the daytime trajectories are similar to those of the corresponding nighttime trajectories.

Averaged diurnal variations of tropospheric columns and near-surface values of aerosols, $NO_2$, $SO_2$, HONO, HCHO, and CHOCHO for the three groups of days are shown in Fig. 12 with the corresponding averaged profiles during the morning (6:00-10:00 BT), around noon (11:00-14:00 BT), and in the afternoon (15:00-18:00 BT). At the bottom of Fig. 12 the averaged diurnal variations of the HONO to $NO_2$ ratio (for both VCDs and near-surface VMRs) and the CHOCHO to HCHO ratio as well as the averaged diurnal variation of the local winds are shown. In order to concisely characterize profile shape, following the procedure in Vlemmix et al. (2015b), the averaged diurnal variations of "characteristic profile heights" $H_{75}$, which is defined as the height below which 75% of the integrated profile resides (75% of the tropospheric column density), is given in Fig.12. The filters applied to the profile inversion results (see Table 2) could systematically impact





results. To avoid drawing a wrong conclusion, the results with and without the filters shown in Fig. 12: in general the effects of the filters are rather small. The results in Fig. 12 indicate that the values of all gas pollutants are the highest for the southerly trajectories, while the aerosol levels are similar for southerly and easterly trajectories. These findings will be discussed in the following in more detail. The lowest values (except for $NO_2$) are found for north-westerly trajectories. These

results are consistent with the findings shown in section 5.1 and 5.2.1. Another interesting finding is the existence of lifted layers of $NO_2$, $SO_2$, and aerosols in the morning for the southerly trajectories indicating the accumulation of pollutants at the top of the boundary layer. This might be caused by the combined effects of higher wind speeds and longer lifetimes at higher altitudes. The averaged $H_{75}$ are ~1.2 km for $NO_2$ and $SO_2$ and ~1.4 km for aerosols, HONO, HCHO and CHOCHO. Systematically lower values of $H_{75}$ for southerly trajectories (high pollution) than for north-westerly trajectories (low

pollution) are found, especially for $NO_2$, $SO_2$ and HONO of down to ~0.5 km in the morning for southerly trajectories. This phenomenon is related to accumulations of anthropogenic pollutants in PBL and PBL dynamics.

   In order to analyse the influence of local winds, bivariate plots of AOD, and the VCDs of $NO_2$, $SO_2$, HONO, HCHO, and CHOCHO VCDs as functions of wind speed and directions are shown in Fig. 13 for the three groups of days, respectively. Note that the data are interpolated to provide a more consistent overview. The original data are also plotted in

Fig. 13 in order to show the representativeness of the interpolated data. In general the highest values of aerosols, $NO_2$, $SO_2$, and HCHO occur for the southeast wind directions on the days of north-westerly trajectories. This finding possibly indicates the effect of short-range transports of pollutants emitted in the downtown area of Xingtai (with an iron factory) about 20km southeast of the station. But of course, also the long range transport might have transported the pollutants via the southeast-local winds to the measurement site (although in general low pollution levels are found for the north-westerly trajectories,

see Fig. 12). The dependences of the pollutants on local winds are not distinct under the two other groups of dominant trajectories. This finding implies two aspects: 1) the Xingtai emissions are not comparable to those from other industrial areas, e.g. near Wuan city; 2) in case that regional transport of pollutants from the south leads to a large-scale pollution in the measurement area, high values of pollutants can be observed no matter where local winds come from.

   Different characteristics are found for the different species in Fig. 12 and 13 and are discussed in the following:

(1) $NO_2$ and $SO_2$

   Similar patterns of diurnal variations of $NO_2$ and $SO_2$ are found in Fig. 12 and 13 and indicate that both species have similar sources and sinks. The increase of $NO_2$ and $SO_2$ before 10:00 BT for days with southerly trajectories (Fig. 12b and c) can probably be attributed to the systematic change of westerly local winds (from the mountain area) to south-easterly winds between 6:00 to 10:00 BT (see Fig. 12I). The same effect could cause the observed high $NO_2$ and $SO_2$ values under

east-southerly wind conditions with a wind speed smaller than 5 m/s, because this wind field frequently occurred around 9:00 BT (see Fig. 12I) when $NO_2$ and $SO_2$ levels are high. The decrease of $NO_2$ and $SO_2$ after around 10:00 BT on the days with the southerly and easterly trajectories (Fig. 12b and c) can be probably attributed to photochemical deposition and dispersion. Dispersion is expected to become more effective when the local wind speeds increase in the afternoon (see Fig. 12i).





(2) HONO

For days with dominant southerly trajectories, high values of HONO were observed, especially in the morning (Fig. 12d). These enhancements are consistent with high levels of $NO_2$ and $SO_2$, indicating that HONO could have been transported during the night because of the absence of photolysis. Under the southerly trajectories, HONO sharply decrease

in the morning and reach the detection limit of about 0.1 ppb near the surface around noon due to its fast photolysis. The photolysis-controlled diurnal variations of HONO also impact the plot of the dependence of HONO on local winds (Fig. 13d) There we can see peak values for light westerly winds. This finding can probably be attributed to the systematically dominant westerly wind in the early morning when HONO values are high. The diurnal variations of the ratios of HONO and $NO_2$ for the three groups of dominant trajectories are also shown in Fig. 12g. Higher HONO / $NO_2$ ratios of up to 6%

were observed in the early morning, especially for southerly trajectories. This finding is consistent with the results shown in Fig. 5 and could be attributed to the faster deposit of HONO than $NO_2$. In addition, for the easterly trajectories, HONO values are higher than for the north-westerly trajectories. This finding is due to high daytime HONO values observed between 6 to 8 June, which are sorted into the group of easterly trajectories. The phenomenon is probably related to a special daytime source of HONO.

(3) HCHO and CHOCHO

Significantly higher HCHO and CHOCHO values for southerly trajectories than for the other trajectories are found in Fig. 12e and f indicating that large proportions of HCHO and CHOCHO sources are related to the transport of anthropogenic emissions from the southwest industrial area. Because of the short life time of HCHO and CHOCHO of on the order of hours, daytime regional transport of HCHO and CHOCHO are probably not significant, but long-lived VOCs can be

expected to be transported and oxidised to HCHO and CHOCHO in the measurement area. In addition, regional transport of HCHO and CHOCHO during night-time could still be possible and explain the high values of HCHO and CHOCHO in the morning. The oxidation of VOCs to HCHO and CHOCHO is probably the dominant source, because the peak values of HCHO and CHOCHO are observed in the late morning and around noon. Correlation coefficients of the near-surface HCHO and CHOCHO VMR with the $O_3$ VMR (measured by an in-situ $O_3$ analyser) are about 0.6 around noon, also indicating a

significant contribution of photo-chemical destruction of VOCs to HCHO and CHOCHO abundances. Moderately high values of HCHO and CHOCHO are found for easterly trajectories indicating that VOC levels in the east area should be considerably higher than in the northwest area. Consistently, the HCHO map derived from the OMI observations in Fig. 2 shows low HCHO values only in the northwest area and comparably high values in all the other areas. Biogenic emissions of VOCs, e.g. isoprene, could also impact the observed HCHO and CHOCHO in May and June. However, the contributions are

not comparable to the anthropogenic sources, because of the low HCHO and CHOCHO observed for north-westerly trajectories. A large forest area is located in the northwest of the measurement area with a distance of about 30 km. The averaged diurnal variations of the ratio CHOCHO/HCHO in Fig. 12h show larger values for southerly trajectories than for other trajectories, especially in the morning. This could be due to different precursor VOCs and different reaction rates regarding the formation of CHOCHO and HCHO. Higher CHOCHO/HCHO ratios are also observed in a rural area near





Guangzhou city in Southern China (Li et al., 2013). Moreover, burning events of residual farm plants could also have contributed to the observed high HCHO and CHOCHO values for southerly and easterly trajectories. This finding will be discussed in more detail in sect 5.3.

(3) Aerosols

Low values of aerosols were observed only for north-westerly trajectories in Fig. 12a. The phenomenon is similar as for HCHO and CHOCHO and also consistent with the similar patterns in the satellite maps of AOD and HCHO shown in Fig. 2. Peak values of aerosols at around 10:00 BT for southerly trajectories are probably due to photochemical formations of secondary aerosols and effects of systematic variations of local winds (as for $NO_2$ and $SO_2$). The profiles of aerosol extinction shown in Fig. 12a indicate that the lifted layers were frequently observed for southerly and easterly trajectories.

The lifted layers could indicate the accumulation and transport of aerosols at high altitudes. In addition burning events of residuals of farm plants could also contribute to aerosols in the measurement area through regional transports. The sources include primary aerosols and secondary aerosols formed in the plume during transports. The effects of burning events will be discussed in more detail in Sect 5.3.

      In general regional transport, especially during night time, is the dominant factor which determines the amounts of

all pollutants in the measurement area. Local winds and photo-chemistries play considerable roles for the corresponding diurnal variations. Wang et al., 2018 demonstrated the same conclusion about aerosols on the significant effect of regional transport based on aircaft measurements operated in the same campaign region and period.

**5.3 Effect of agricultural burning**

      There are extensive farmlands in the NCP region, and farmers normally burn residuals of plants after harvest,

especially wheat straw in May and June. Burning events during the measurement period are identified from the Fire Information for Resource Management System (FIRMS) based on MODIS satellite observations (https://earthdata.nasa.gov/earth-observation-data/near-real-time/citation#ed-firms- citation). The daily total numbers of fire points in the NCP region during the campaign, Fig. 14a shows frequent burning events occurred on 10, 15, 19, 24, 29 May and 4, 9 June. A Map of all observed fires during the entire measurement period (Fig. 14b) indicates that burning events

mostly appeared in the east and south of the measurement site. Burning impacts air quality on days with southerly and easterly trajectories.

      Indeed, high values of HCHO, CHOCHO, and aerosols can be seen in Fig. 5 on most days with high numbers of burning events and also one or two days after the events. In contrast, on 15 and 24 May, when the trajectories originated mainly from north-westerly directions, no enhanced levels of HCHO, CHOCHO and aerosols are found. Therefore we can

expect that VOCs and aerosols emitted from the burning plants were transported to the measurement area and considerably impacted the abundances of HCHO, CHOCHO, and aerosols (aerosols might also be additionally formed from the photochemical degradation of the VOCs) for southerly and easterly trajectories. VOCs and aerosols emitted from burning events could still impact the measurement area in two days after the events because of their long atmospheric lifetimes. Here




it is important to note that although $NO_x$ is probably also emitted from the biomass burning events, it might be mostly destroyed during the transport to the measurement site because of its short life time.

One interesting example is found on 6 June for dominant south-easterly trajectories. Peak values of HCHO, CHOCHO, aerosols, and ozone are found in the afternoon on 6 June (see Fig. 5 and Fig. S3 in the supplement). However the $NO_2$ and $SO_2$ values are quite low. The true colour images of the area observed by MODIS from 4 to 6 June are shown in Fig. 14c indicating that the whole NCP region is partially cloudy and covered by dense haze with an AOD of up to 2 at 550 nm, but almost no fire points are identified on 6 June. The AOD is derived from the MODIS aerosol product supplied by NASA (http://ladsweb.nascom.nasa.gov/data/search.html). Many fire points are observed on 4 June, when the aerosol load is low. Although no fire points can be seen on 5 June, this doesn't necessarily indicate there are no fires below the clouds because clouds shield satellite observations. Therefore VOCs emitted from the burning events on 4 June (probably also on 5 June) could strongly contribute to the peak values of HCHO, CHOCHO, aerosols on 6 June through photo-chemical reactions. Effective photo-chemical reactions are expected on 6 June due to the cloud free conditions and are also implied by the high ozone values in the afternoon. Also primary aerosols and secondary nitrate might contribute to the observed high aerosols on 6 June. Consistently, the collocated measurements of the aerosol composition reported in Zhang et al. (2018) demonstrated that the high amounts of aerosols are dominated by organic aerosols (about 40 $\mu g\ cm^{-3}$ on 6 June), and also have high sulphate and nitrate fractions (about 20 and 15 $\mu g\ cm^{-3}$). The observed phenomenon of relatively high values of HCHO, CHOCHO and aerosols, but low values of $NO_2$ and $SO_2$ for easterly trajectories could thus at least partly be due to effects of burning events.

## 5.4 Rough estimates of contributions of regional transports

The local emissions (including the contributions of local transport from the downtown area of Xingtai city) can be treated as the dominant sources of pollutants for situations with transport from the north-west (the group of days with north-westerly trajectories). The contributions of regional transport to the observed pollutants for the two other groups of days can be roughly estimated based on the relative differences of the pollutant values compared to those for north-westerly trajectories. Tropospheric columns are used for the estimation because regional transport often occurs not directly above the surface. According to this simple calculation, for the days with mainly southerly trajectories, about 47%, 45%, 47%, 34%, 46% and 65% of the observed amounts of $NO_2$, $SO_2$, HONO, HCHO, CHOCHO, and aerosols, can be assigned to the effect of regional transport, respectively. In summary, we find that the total contribution of regional transport to the total amounts of pollutants in the measurement area during the entire measurement period is about 29%, 25%, 27%, 22%, 28%, and 54% for $NO_2$, $SO_2$, HONO, HCHO, CHOCHO, and aerosols, respectively. It needs to be clarified that these results are only rough estimates. The uncertainties largely depend on the uncertainties of the trajectories, variations of the local emissions and chemical reactions. The error budget cannot be well constrained at this stage.





## 6. Conclusions

Vertical profiles, near-surface, and column densities of aerosol extinction, $NO_2$, $SO_2$, HONO, HCHO, and CHOCHO were retrieved from MAX-DOAS measurements during the period from 8 May to 10 June 2016, at a rural site located on the central-west edge of the NCP. The mean value of near-surface aerosol extinction was ~0.43 km$^{-1}$, with high

levels of gaseous pollutants $NO_2$, (7.8 ppb), $SO_2$, (17.1 ppb), HONO (0.22 ppb), HCHO (3.3 ppb), and CHOCHO (0.08 ppb). The mean value of AOD at 360 nm was ~0.65, with high VCDs of gaseous pollutants $NO_2$ (15.6 $\times 10^{15}$ molecules cm$^{-2}$), $SO_2$ (31.8 $\times 10^{15}$ molecules cm$^{-2}$), HONO (0.22 $\times 10^{15}$ molecules cm$^{-2}$), and HCHO (13.8 $\times 10^{15}$ molecules cm$^{-2}$). The HONO/$NO_2$ ratios averaged 1.9% to 2.9% with a peak of about 5% in the morning on days with transport of air from polluted south or southwest areas. CHOCHO/HCHO ratios averaged between 2% to 2.5% with a peak of about 3.5% in the

morning on days with transport of air from the polluted areas. Significant day-to-day variations were found for all species mainly due to regional transport of pollutants and changes in synoptic patterns. Agricultural burning events impacted considerably the day-to-day variations of HCHO, CHOCHO, and aerosols. Maximum values systematically occurred in the morning for $NO_2$, $SO_2$, and HONO, but around noon for aerosols, HCHO, and CHOCHO. The diurnal variations were dominated by characteristic photochemistry, upslope/downslope circulation, and PBL dynamics. Aerosols, HCHO, and

CHOCHO profiles with $H_{75}$ of ~1.4 km typically extended to higher altitudes than $NO_2$, $SO_2$, and HONO with $H_{75}$ as low as 0.5 km under polluted condition, probably due to secondary formation. Lifted layers were systematically observed for all species (except HONO), indicating accumulation, secondary formation, or long-range transport of the pollutants at high altitudes. At high altitudes, pollutants have longer lifetimes and winds are stronger, leading to large-scale adverse impacts.

AOD (R of 0.84, slope of 1.13) and near-surface aerosol extinction (R of 0.81, slope of 1.02) derived from MAX-

DOAS measurements were consistent with sun-photometer measurements and visibility-meter measurements under cloud free conditions with low aerosol loads. Near-surface VMRs of $NO_2$, $SO_2$, and HCHO were well correlated with in-situ measurements with R of 0.9, 0.95, and 0.6, respectively. However, MAX-DOAS results are considerably smaller than the in-situ results mainly due to vertical and horizontal inhomogeneities of trace gases. In general, the agreement of all species between MAX-DOAS and other measurements weakens under cloudy and high aerosol conditions. We further compare

profiles of aerosol extinction retrieved from the MAX-DOAS with the Lidar measurements on two days of simultaneous measurements. Reasonable consistency, but also systematic differences are found, which were mainly caused by differences in sensitivity as a function of altitude and substantial horizontal gradients of aerosols. Also cloud contamination of MAX-DOAS results is probable for some measurements. Vertical profiles of aerosol extinction, $NO_2$ and $SO_2$ VMRs retrieved from MAX-DOAS measurements are also compared with the aircraft measurements on two days, and generally indicate

reasonable consistency, after the MAX-DOAS averaging kernels are applied to the aircraft data and vertical profiles are extrapolated to observed surface values. The smoothing effect can cause MAX-DOAS retrievals to underestimate pollutants above 2 km and overestimate below.



We analysed the effects of regional and local transport of pollutants based on case studies and a systematic analysis using the MAX-DOAS measurements, backward trajectories, synoptic situations, and local winds. In general, the regional transport, especially during nighttime, is found to be the dominant factor in local air quality. For surface values, local winds, photochemistry, and PBL dynamics all exert a strong influence on the diurnal variation of the pollutants. The regional transport of gas pollutants plays a more significant role during night time than daytime due to longer life times at night. We document regular episodes of regional transport of clean air masses from the north-west (often associated with a cold front), and polluted air masses from the southern industrialized areas around Wuan city with many steel and coal coking facilities. Burning events of crop residuals in the NCP region can considerably impact HCHO, CHOCHO, and aerosols. Contributions of regional transport to the total amounts of pollutants in the measurement area during the entire measurement period were 20% to 30% for trace gases, and about 50% for aerosols.

**Author contribution:** Yang Wang[1] analysed vertical profiles and regional transports of pollutants by combing different data sets and trajectory simulations, and prepared the manuscript with contributions from all co-authors. Yang Wang[1], Steffen Dörner, Sebastian Donner, and Thomas Wagner designed, operated and analysed the MAX-DOAS measurements. Sebastian Böhnke contributed to the analysis of regional transports in Sect 5.2.1. Yang Wang[4] contributed to the operation of MAX-DOAS measurements. Hao He, Xinrong Ren, and Russell R. Dickerson operated and analysed the aircraft measurements. Zipeng Dong, Dong Liu, Zhenzhu Wang, and Jiwei Xu operated and analysed Lidar measurements. Zhengqiang Li, Donghui Li, and Hua Xu operated and analysed sun-photometer measurements. Yuying Wang operated in-situ measurements. Isabelle De Smedt and Nicolas Theys contributed the OMI satellite data of HCHO and SO$_2$. Zhanqing Li designed and organized the A$^2$BC campaign.

**Acknowledgements:** We thank the Wuxi CAS Photonics Co. Ltd for their contributions to operate the observations of the MAX-DOAS instrument, the long path DOAS instrument, the visibility meter and the weather station in Wuxi. We thank the Institute of Remote Sensing / Institute of Environmental Physics, University of Bremen, Bremen, Germany for their freely accessible radiative transfer model SCIATRAN. We thank Belgian Institute for Space Aeronomy (BIRA-IASB), Brussels, Belgium for their freely accessible WINDOAS software and generating the mean map of SO$_2$ and HCHO tropospheric VCDs derived from OMI observations over eastern China. We thank Royal Netherlands Meteorological Institute for their freely accessible archive of OMI tropospheric NO$_2$ data. We thank Steffen Beirle in MAX Planck institute for Chemistry for his help on review the paper innerly. We thank Prof. Tong Zhu and his group in Peking University for their sharing an ECOTECH instrument for the campaign. The US National Science Foundation (Grant #1558259) supported ARIAs.



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





# Figures

Figure 1: Topography maps (http://en-au.topographic-map.com/places/China-8955742/) and satellite images (Google map) of the region around the measurement station (37.18 °N, 114.37 °E), marked in subplots (a) and (b) by black dots. (a) Topography map in longitude and latitude ranges of ±4 °(about ±360 km) around the measurement station; colours indicate the terrain height; a map indicating the location of the selected region in China is shown in the top-left corner. (b) Topography map of the area marked by the red square in (a); the blue and red squares indicate the areas of Xingtai and Wuan; the blue (NW wind) and red (SE wind) arrows represent the dominant wind directions before and after 10:00 BT (GMT +8), respectively. (c) Satellite image of the area around Wuan marked by the red square in (b); an





industrial area (with many stacks of several iron and coal coking facilities) and the downtown area of Wuan are located in the northwest and southeast parts of the image. (d) Zoomed satellite image of one iron and coal coking units marked by the red square in (c).



5    Figure 2: Maps of average tropospheric VCDs of (a) NO$_2$ from DOMINO v2 (Boersma et al., 2007 and 2011), (b) SO$_2$ (Theys et al., 2015) and (c) HCHO (De Smedt et al., 2008, 2012 and 2015) from BIRA, derived from OMI observations. Average AODs derived from the

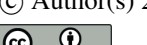


MODIS AQUA instrument is shown in (d). The maps show the same area over Eastern China as shown in Fig. 1a). Data are averaged for May and June of all years from 2012 to 2016. The black dots indicate the measurement station. The grey squares mark the same area as shown in Fig. 1b.

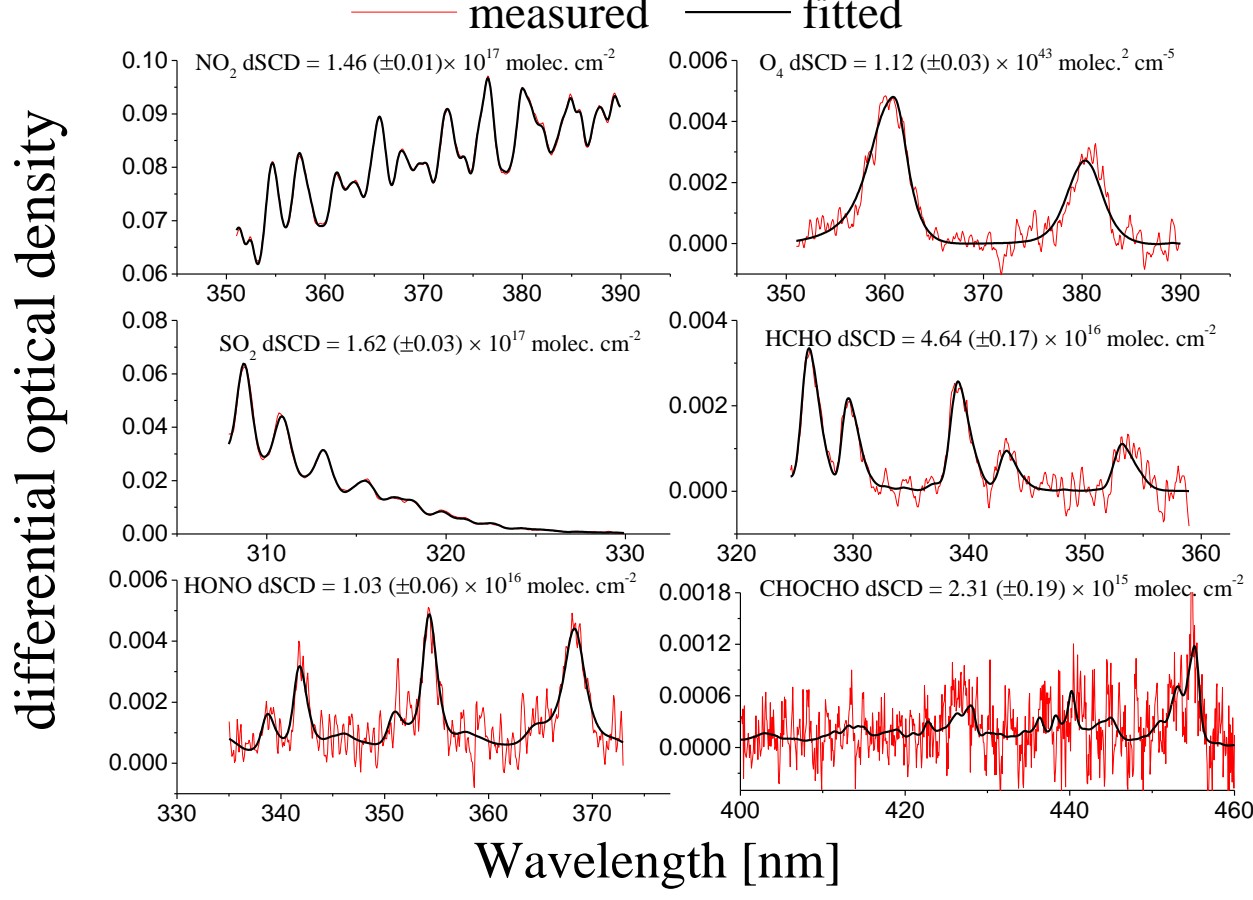

Figure 3: Examples of typical DOAS fits of $NO_2$, $O_4$, $SO_2$, HCHO, and HONO around noon on 17 May, and CHOCHO on 27 May 2016. The black and red curves indicate the fitted absorption structures and the derived absorption structures from the measured spectra, respectively. The fitted dSCDs (and fit errors in brackets) are given in the individual subfigures.







Figure 4: Examples of results derived from MAX-DOAS measurements on 11 May 2016 (with high pollution levels). (a) Cloud classification results. (b) Time series of vertical profiles of aerosol extinction (including cloud contaminated results), $NO_2$, $SO_2$, HONO, HCHO, and CHOCHO VMRs. The black rhombuses (diamonds) plotted above individual subfigures mark 'high confidence profiles' (results with deviations between measured and simulated dSCDs smaller than the individual thresholds, see Table 2. (c) Averaging Kernels (AK) of profile retrievals around noon with information on the degree of freedom (ds). (d) a-priori profiles and retrieved profiles (with corresponding VCDs) around noon.





Figure 5: Daily averaged (for three different periods of the day) near-surface aerosol extinction and trace gas VMRs (left), and AODs and trace gas VCDs (right) for the whole campaign. The blue, red, and green colours indicate results for the time periods of 6h-10h, 10h-14h, and 14h-18h, respectively. The ratios of HONO versus NO₂ and CHOCHO versus HCHO are also given for the near-surface VMRs (left) and VCDs (right). The colour coded numbers in the brackets on the top of each subfigure give the averaged values for the different daily periods.

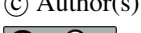



(a)

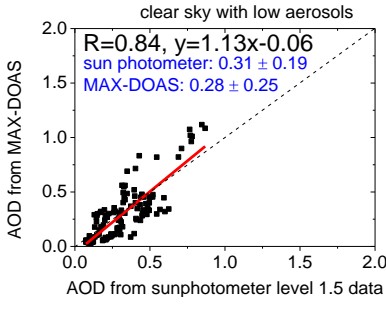
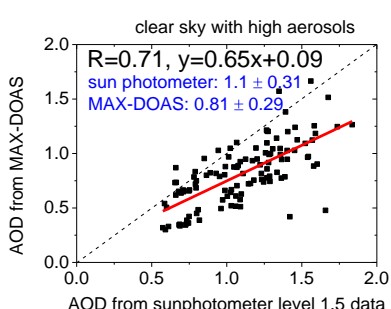

(b)

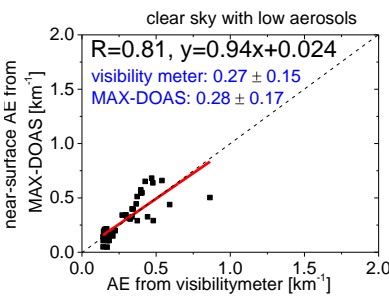
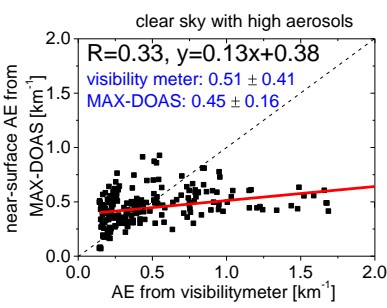
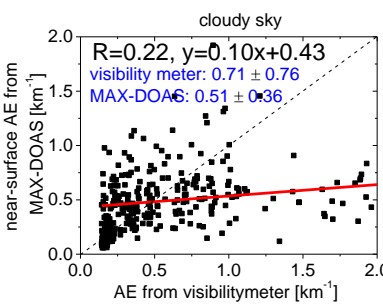

5    (c)

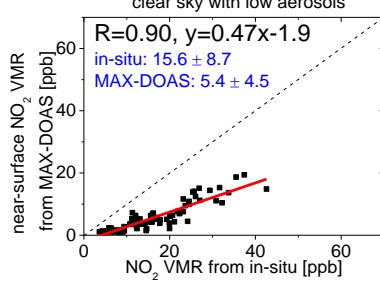
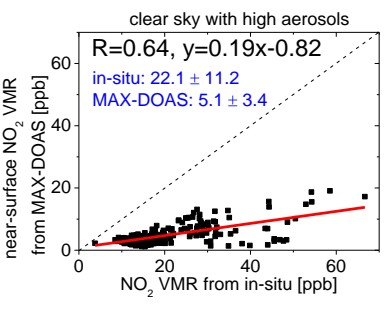
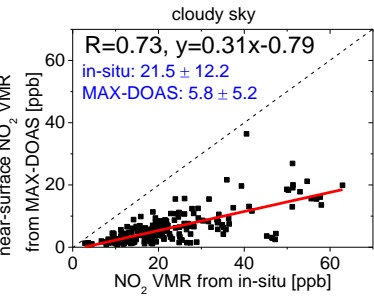

(d)

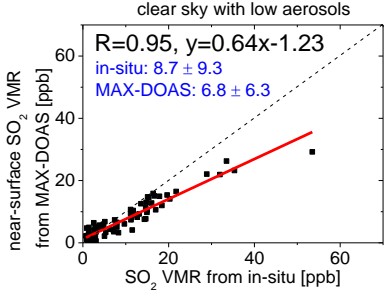
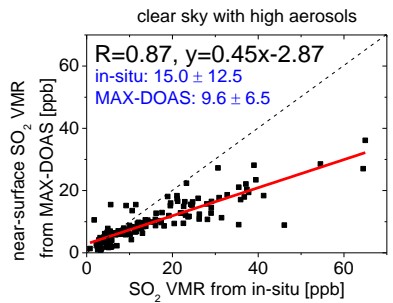
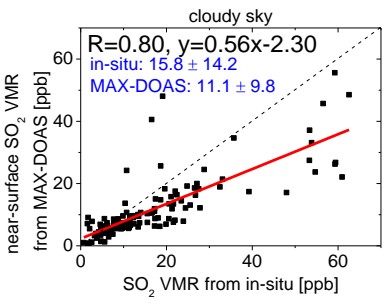





(e)

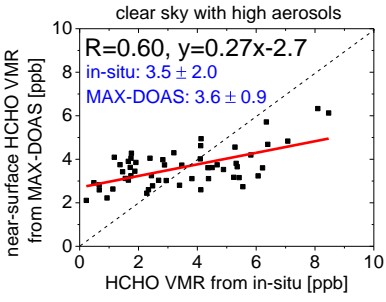 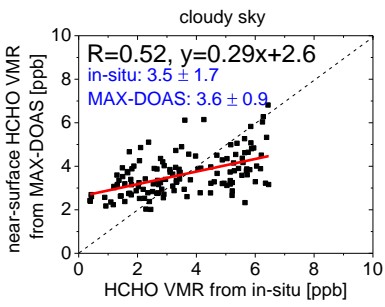

Figure 6: Correlation plots of AODs (a), near-surface aerosol extinction (b), and VMRs of $NO_2$ (c), $SO_2$ (d), and HCHO (e) derived from MAX-DOAS measurements versus results from other techniques. Mean values and standard deviations of the compared values are marked by the blue colour and given before and after '±' in individual subfigures. AODs are taken from the Level 1.5 product (cloud-screened and quality controlled) derived from sun photometer measurements. Note that the Level 2.0 (quality-assured) AOD product is not (yet) available. Near-surface aerosol extinction is derived from visibility meter measurements. Surface VMRs of $NO_2$, $SO_2$, and HCHO are derived from the in-situ instruments (see section 3).

(a)

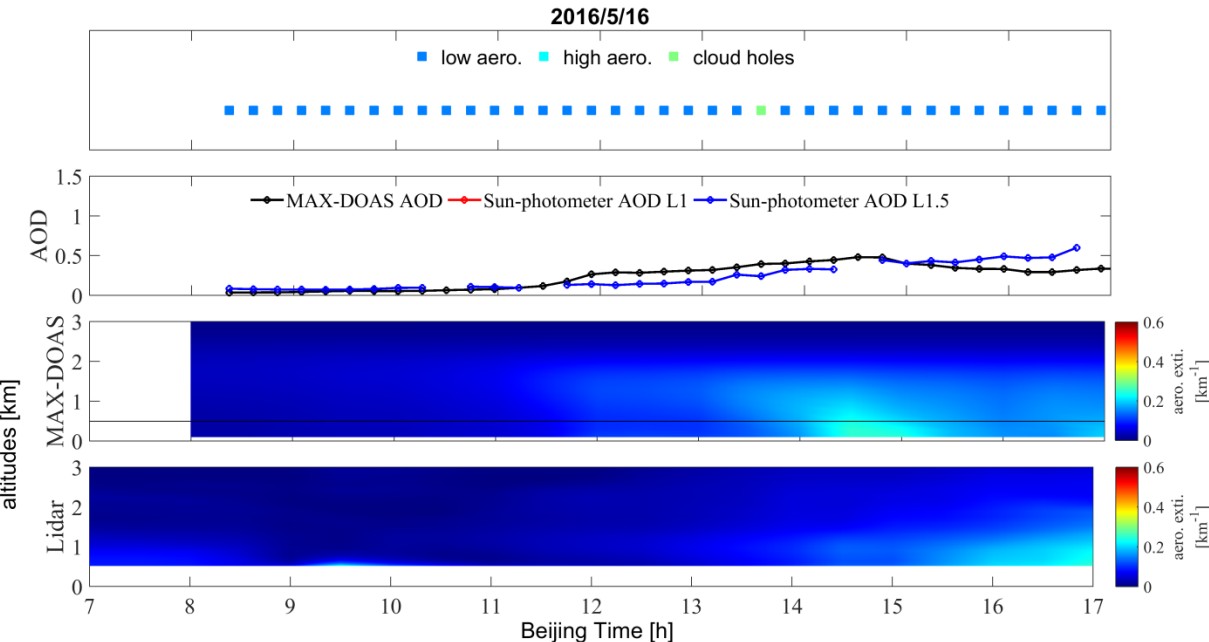



(b)

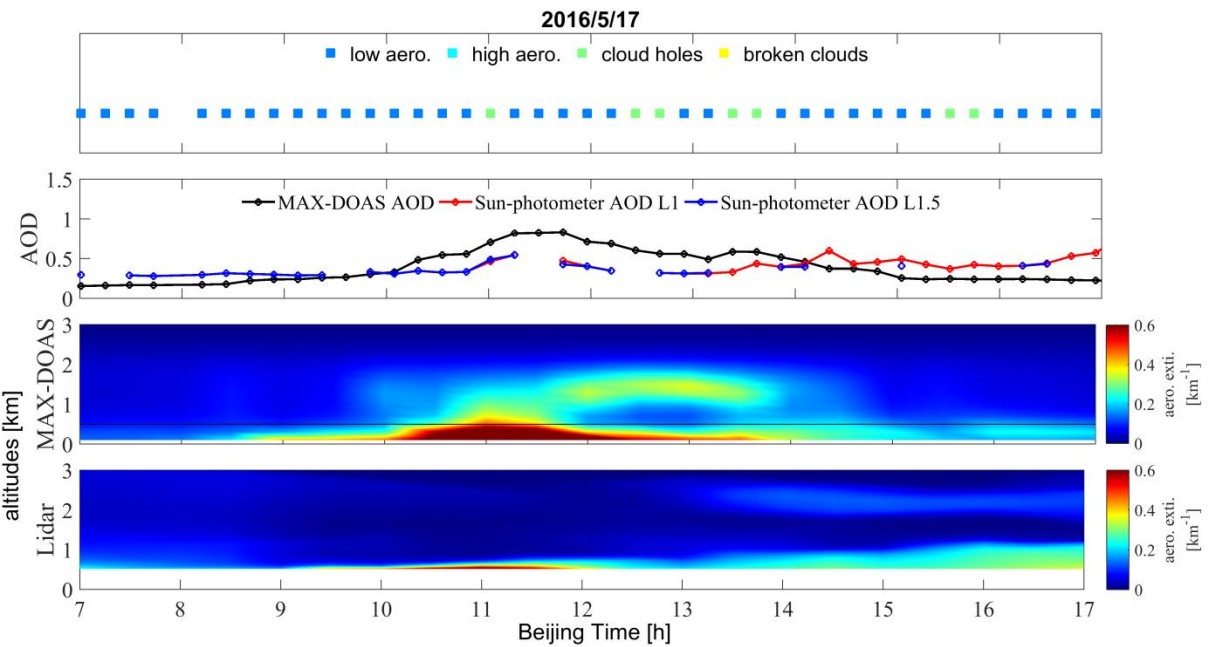

Figure 7: Vertical profiles of aerosol extinction derived from MAX-DOAS and Lidar measurements on 16 (a) and 17 (b) May 2016. Cloud classification results are shown at the top of both subfigures. AODs (black) are derived from the MAX-DOAS measurements and from the sun photometer. The red and blue lines shown in the second row are the level 1 (unscreened) and level 1.5 (cloud-screened and quality controlled) products, respectively. The black line in the third panel marks the lower limit (500m) of the LIDAR profiles.

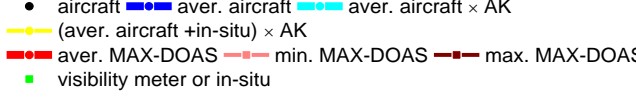

(a) 8 May, 2016

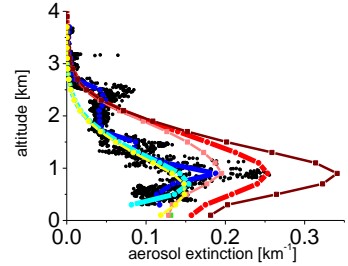
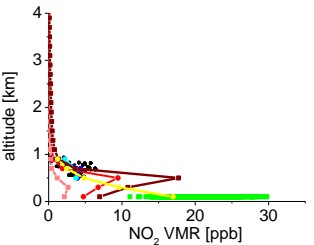
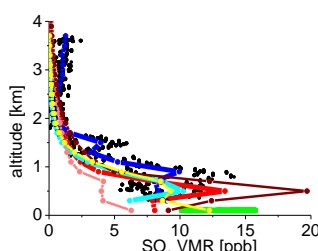



(b) 21 May, 2016

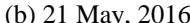

Figure 8: Vertical profiles of aerosol extinction, and VMRs of NO$_2$ and SO$_2$ derived from the MAX-DOAS measurements compared with corresponding aircraft measurements on 8 (a) and 21 (b) May 2016. The black dots represent the original aircraft measurements. The light and dark blue curves show the averaged and smoothed (with the averaging kernels) aircraft data. The red, pink and brown lines indicate the averaged, maximum, and minimum profiles derived from the MAX-DOAS measurements during the overpass time of the aircraft measurements. The near-surface values from the visibility meter and in situ trace gas measurements are indicated by the green dots. The yellow curves show the averaged and smoothed combined profiles from aircraft and surface measurements.

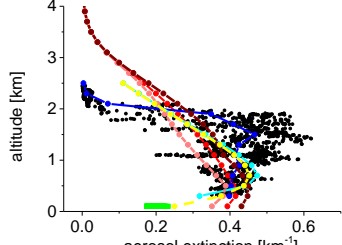
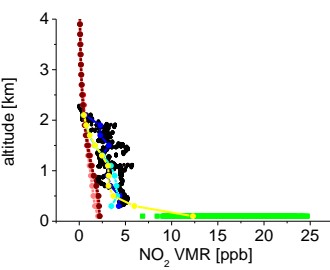
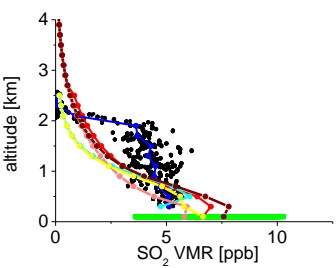

Figure 9: Photos taken by a camera along the line of sight of the MAX-DOAS instrument in the period from 11 to 16 May (right), 2016.





Figure 10: Results from MAX-DOAS measurements, trajectories, meteorological data, and independent measurements of pollutants during the period from 11-16 May 2016. The figures surrounded by the red and blue dashed boxes show the values for 24-hr periods (red) or 12-hr (blue) daylight periods (6h-18h). The total fire points in the NCP area are derived from MODIS observations and the hourly accumulated precipitation was measured at the station (third row). In the fifth row hourly averaged local winds are given by arrows moving with the wind. The colour bars of the MAX-DOAS profiles are given in the bottom. The black rhombuses plotted above the individual subfigures of the time series of MAX-DOAS profiles mark most reliable (thresholds are given in Table 2).







## (c) HCHO VCD

## (d) AOD



Figure 11: Average reproduced maps (inside the red dashed square) of column densities of $NO_2$ (a), $SO_2$ (b), HCHO (c), and AOD (d) based on MAX-DOAS measurements and back-trajectories with different backward times (for details see text), and comparison with maps of the pollutants derived from OMI and MODIS satellite observations. Different times of the backward trajectories are used for the generation of the maps. The correlation coefficients of the pollutants in the reproduced maps against the satellite map are given in individual subfigures. Note that 1 °latitude and longitude are about 90 km.








Figure 12: Averaged MAX-DOAS results for the three groups of days with different dominant directions of nighttime trajectories (different colors). In the left two columns of subfigures (a) to (f), the diurnal variation of AOD, trace gas VCDs, near-surface aerosol extinction, and near-surface trace gas VMRs are shown (circles and stars indicate the filtered and unfiltered results based on the deviations of measured and modelled dSCD in the profile inversion). In the middle column of subfigures, the diurnal variations of $H_{75}$ of retrieved profiles are shown. In the right two columns of subfigures the corresponding averaged profiles for three periods (different line styles) are shown. The subfigures (g, h, i) show the averaged ratios of HONO to $NO_2$ VCD and near-surface VMRs, the averaged ratios of CHOCHO to HCHO VCDs and near-surface VMRs, and the averaged diurnal variation of the local wind fields, respectively. In general, pollutant levels are the highest with winds out of the south and east where major sources reside and lowest with winds out of the north or northwest generally behind cold fronts. HONO/$NO_2$ ratios are the highest for winds from the east, where the agricultural activity is and CHOCHO/HCHO ratios are the highest for back trajectories out of the south, indicating the dominance of petrochemical activity there.









Figure 13: Bivariate figures of AOD (a) and VCD of $NO_2$ (b), $SO_2$ (c), HONO (d), HCHO (e), and CHOCHO (f) as function of wind speed and direction for the three groups of days with the different dominant directions of nighttime trajectories. The colours show values of AOD and VCD.

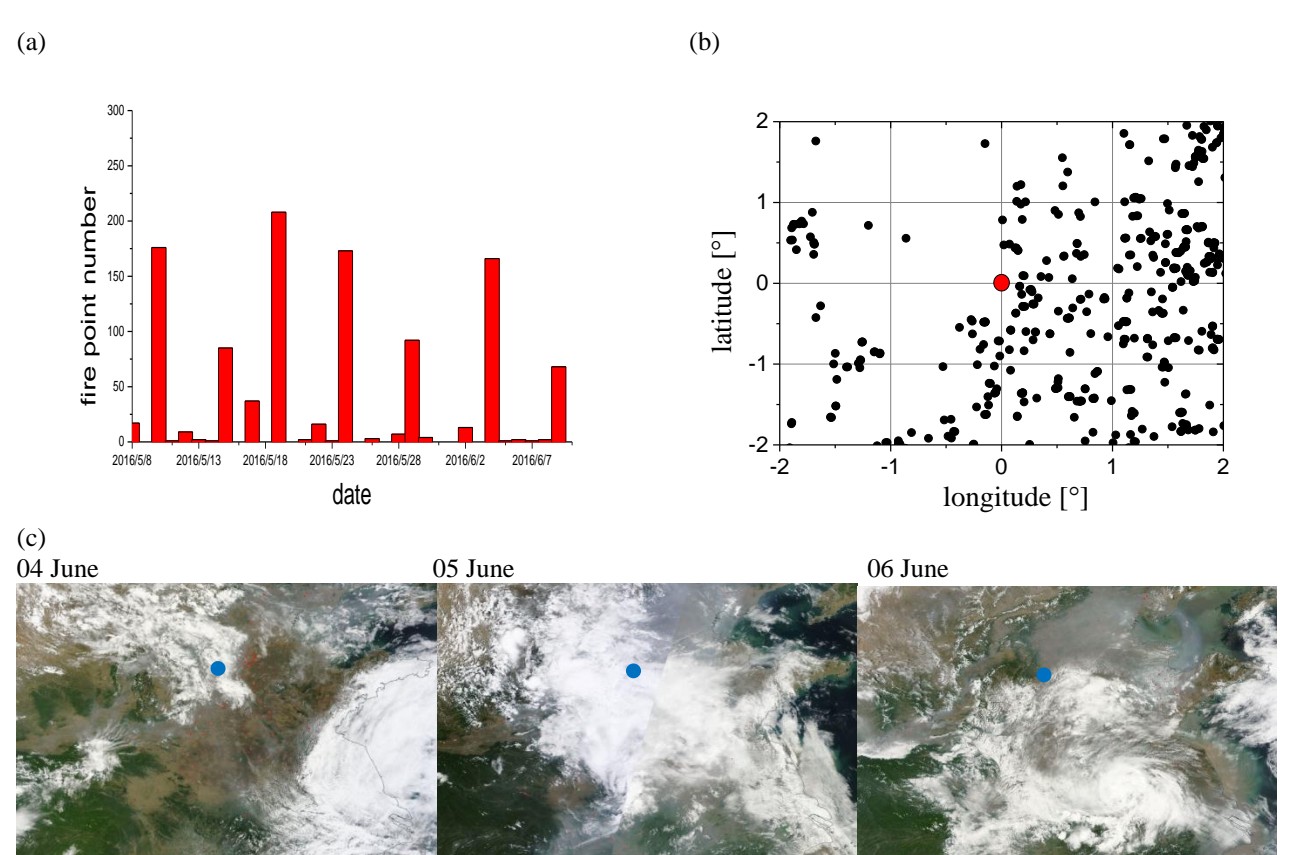

Figure 14: (a) daily total numbers of fire counts in the NCP region. (b) Distributions of fire points in the $2° \times 2°(180 \text{ km}^2)$ area around the measurement site. (c) Images of the NCP region derived from MODIS observations; blue circles and red dots represent the measurement station and fire points, respectively.





## Tables

Table 1 Settings used for the $O_4$, $NO_2$, $SO_2$, HCHO, and CHOCHO DOAS analyses. Note that the settings for $O_4$, $NO_2$, $SO_2$, HCHO follow the suggestion in Wang et al., 2017a. The settings for HONO and CHOCHO follow the suggestions in Wang et al., 2017c and the MAD-CAT campaign (Ortega et al., 2015), respectively.

| Parameter | Source | Species | | | | | |
|---|---|---|---|---|---|---|---|
| | | $O_4$ | $NO_2$ | $SO_2$ | HCHO | HONO | CHOCHO |
| Fitting spectral range | | 351-390nm | 351-390nm | 307.8-330nm | 324.6-359nm | 335–373nm | 400–460nm |
| Cross section | $NO_2$: Vandaele et al. (1998), 220 K, 298 K | × | × | ×(only 298 K) | ×(only 298 K), $I_0$-corrected[*] ($10^{17}$ molecules/cm$^2$) | ×I0-corrected* ($10^{17}$ molecules cm$^{-2}$); Taylor terms[#] with respect to $\sigma_{NO_2}$ at 298 K : $\lambda\sigma_{NO_2}$, $\sigma_{NO_2}^2$ | ×I0-corrected* ($10^{17}$ molecules cm$^{-2}$); Taylor terms[#] with respect to $\sigma_{NO_2}$ at 298 K : $\lambda\sigma_{NO_2}$, $\sigma_{NO_2}^2$ |
| | $O_3$: Bogumil et al., (2003), 223 K and 243 K | ×(only 223 K) | ×(only 223 K) | × | ×(only 223 K) $I_0$-corrected[*] ($10^{18}$ molecules cm$^{-2}$) | ×$I_0$-corrected[*] ($10^{18}$ molecules cm$^{-2}$) | ×(only 223 K) |
| | $O_4$: Thalman and Volkamer (2013), 293 K | × | × | × | × | × | × |
| | $SO_2$:Bogumil et al. (2003), 293 K | | | × | × | | |
| | HCHO: Meller and Moortgat (2000), 293 K | × | × | × | × | × | |
| | HONO: Stutz et al. (2000), 296 K | | | | | × | |
| | $H_2O$ (vapor): Polyansky et al. (2018) | | | | | × | × |
| Ring | Ring spectrum calculated from DOASIS (Kraus, 2006) and additional Ring multiplied by $\lambda^{-4}$ (Wagner et al., 2009) | × | × | × | × | × | × |
| Polynomial degree | | 3 | 3 | 5 | 5 | 5 | 5 |



| Intensity offset | | constant | constant | constant | constant | Polynomial of order 1 (corresponding to 2 coefficients) | Polynomial of order 1 (corresponding to 2 coefficients) |
|---|---|---|---|---|---|---|---|

\* solar $I_0$ correction, Aliwell et al., 2002.
\# Pukīte et al. 2010.

5   Table 2 Different filters and corresponding thresholds applied to the MAX-DOAS results. The thresholds are experientially determined to exclude most of outliners. Also the corresponding fractions of remaining data are indicated. (SZA: solar zenith angle; RIO: relative intensity offset in the DOAS fit; RMS: root mean square of the residual in the DOAS fit)

| | Aerosols ($O_4$) | $NO_2$ | $SO_2$ | HCHO | HONO | CHOCHO |
|---|---|---|---|---|---|---|
| SZA | < 85 ° | < 85 ° | < 70 ° | < 85 ° | < 85 ° | < 85 ° |
| RIO | < ±1% | <±1% | <±1% | <±1% | <±1% | <±1% |
| RMS | < 0.002 | < 0.002 | <0.01 | <0.002 | <0.0012 | <0.001 |
| deviations of modelled and measured dSCD | < $2.5 \times 10^{42}$ molecules$^2$ cm$^{-5}$ | <$1 \times 10^{16}$ molecules cm$^{-2}$ | <$1 \times 10^{16}$ molecules cm$^{-2}$ | <$1 \times 10^{16}$ molecules cm$^{-2}$ | <$0.8 \times 10^{15}$ molecules cm$^{-2}$ | <$0.4 \times 10^{15}$ molecules cm$^{-2}$ |
| Sky condition | Excluding all cloudy conditions | Excluding data under thick cloudy conditions | | | | |
| Remaining percentage | AOD and profiles:33% Near-surface:86% | 62% | 58% | 68% | 57% | 56% |

