# Peer review of "Vertical profiles of NO2, SO2, HONO, HCHO, CHOCHO, and aerosols derived from MAX-DOAS measurements at a rural site in the central-western North China Plain and their relation to emission sources and effects of regional transport"

_Atmospheric Chemistry and Physics, 2018_

## Referee Comment (RC1) · Anonymous Referee #1 · 30 Dec 2018

**General comments**

Yang Wang et al. presented a comprehensive study of the temporal and spatial distribution of aerosols and trace gases in the central-western North China Plain. The manuscript is well structured and the results show good correlation with results of other instruments.

However, many results were filtered based on a cloud classification scheme which performs somehow unreliable to me. Especially low and high aerosol loads seem to be mixed up with clouds. To a certain points, MAX-DOAS profiles should be able to retrieve these differences but do not show larger deviations e.g. on 11/05 and 19/05. Since this classification has a large impact on the complete discussion, I would suggest to add a small estimate of the impact of wrongly classified/filtered scenarios on your results.

Furthermore, an additional analysis of NO2 retrieved in a different fitting window might help to clarify if horizontal inhomogeneties were present.

**Specific comments**

**P2, L14-19:** Please add a reference to Fig 1. in the Introduction.

**P2, L32:** Please add the full name for the abbreviation East-Aire.

**P3, L9:** MAX-DOAS algorithms are not only based on OE. Iterative approaches like Newton-Gauß or Levenberg-Marquardt are in use. I would rather call these algorithms "inversion algorithms" or "inversion based algorithms".

**P5, L17:** Wang et al 2018 seems to be the wrong reference?

**P5, L26:** "and for the MAD-CAT campagin" → "e.g. the MAD-CAT campaign"

**P5, L30:** You state detection limits but not how they were calculated. Please add the missing information.

**Section 2.2.3:** The uncertainties of the individiual parameters were given as percentages but where do they come from and how were they calculated? From a previous study or from not shown sensitivity tests? Please give some information.

**P6, L11-12:** When there is a sunphotometer measuring routinely. Why not using the exact SSA and asym parameter closest in time rather than averaged values? For which wavelengths are the averaged quantities? How did you convert to the proper wavelengths or did you assumed no wavelength dependency?

**P6, L17:** How were these wavelengths chosen? 354 for HONO is the mid of the fitting window but what about the other wavelengths? Was there are reason for not choosing the fitting window mid wavelengths?

**P6, L21:** Why was the upper grid limit chosen to be at 3km? Typical altitudes from other studies are usually at 4km.

**P6, L25:** Covariances of 100% of the surface value for all altitudes? Is this correct? The commonly used approach is a fixed percentage of the a priori profile of the individual altitudes.

**Table 2:** Why is there a different SZA limit for SO2 compared to the other trace gases?

**P6, L30:** Here, you write R but in the Figure its R^2. Which one was given?

**Fig 4:**

Please change the colors for either low or high aerosols in this and similar plots because it is hard to distinguish between both markers.

1. The aerosol retrieval shows similar profile shapes from 6 to 13 but the cloud classification finds different cloudy conditions, sometimes with thick clouds. How is it possible that the aerosol retrieval is not affected by thick clouds?

2. The a priori profiles for aerosols and SO2 are not even close to the the retrieved profiles. How can you be sure that you do not over- or underestimate the retrieved profiles due to in inaccurate a priori profile?

3. I do not understand why the degrees of freedom for aerosols are larger than any of the

trace gas ds. This is unexpected for me. Could you please explain where these larger differences in ds for the invididual retrievals come from?

**P7, L15:** Why is the near-surface extinction trusted under partially cloudy conditions but the AOD not? I would also assumed that the near-surface extinction is inaccurate when broken clouds led to a contaminateion of some elevation angles only. The profile will be smoothed due to the a priori smoothing and the retrieval for all altitudes should be affected.

**P7, L25-28:** The numbers are averages over the full time period? Please call it a total average then or specify what these numbers exactly mean.

**P8, L3-5:** 5-times the word "also" in three lines. Maybe you can reformulate these two sentences.

**P10, L6-7:** I am confused of what you wrote in Section 2.2.3 and what you wrote in lines 6-7 (see comment P6, L11-12). Please explain your approach in greater detail.

**P10, L10:** A parameter of 1 means an Angström Exponent of 1? Please replace the word "parameter". Furthermore, was the exponent of 1 used for all data or was the sunphotometer's Angtröm exponent closest in time applied for the conversion?

**P11, L28-29:** In addition to your reason, a general lower sensitivity for higher altitudes might be another reason. Furthermore, the limitation to 3km might also be an issue.

**P12, L1-2:** Generally, I would not assume that an elevated layer within the lowest kilometre can not be resolved by a MAX-DOAS profiling algorithm. For higher altitudes, this might be an issue. Is it possible to add a brief synthetic test on 3 or 4 elevated layers in different altitudes? Just to see the retrieval response. Because an elevated layer could also be possible when die underlying aerosol profile is box-like due to an oscillation around this box-features.

**Fig 6:**
  1. It would be interesting to know if these outiers for high visibilitymeter AE correspond to certain geometries, time or weather conditions?
  2. **P12, L11:** When this behaviour for different cloudy conditions can be attributed to clouds, why is there a different correlation for NO2 and SO2 for high aerosols and cloudy sky? NO2 has a better correlation for cloudy sky while SO2 has a better correlation for high aerosols. Do you expect large inaccuracies in the classification scheme for clouds and aerosols?

**P12, L12-14:** A limited vertical sensitivity for near-surface trace gas concentrations might also be important when the trace gas is concentrated in a shallow layer much smaller than the grid step width of the profiling algorithm, especially when using a coarse grid steps width of 200m.

**Section 4.2:**
  1. What is the vertical resolution of the Lidar?
  2. Was AK smoothing applied for the Lidar measurements (with the assumption that the vertical resolution is higher than that for MAX-DOAS)? Please add also the Lidar AOD to the AOD sub-figure.
  3. Why are high near-surface values not similarly found by both instruments? The sensitivity for near-surface values should be the highest for both instruments and different air masses and clouds are not expected to be that important for lower altitudes. E.g. MAX-DOAS found larger extinctions at 15:00 (2016/5/16) while the Lidar found larger values in the evening of 2016/5/17.

**P13, L8:** How did you calculate these "combined profiles"? Linear interpolation between lowest air-craft and surface value? Please add some information.

**Fig 8:**
  1. Please add times for MAX-DOAS measurements and the overpasses. It would also be nice to have some color-coding or different grey-scales for the green surface values to identify the surface value changes throughout the measurement period.
  2. Why do the SO2 curves always agree better than NO2 even though the degrees of freedom are much lower for SO2? Did you try to retrieve NO2 also in another fitting window (> 400nm)? It would be interesting to see if the results differ strongly. That would support the argument of horizontal inhomogeneties.

**P13, L24-25:** Please add the time when the photos were taken on the individual days.

**Section 5.1:** Why does the cloud classification show highly variable results between 6 and 14 BT (11/05) but the aerosol profiles do not differ strongly? This indicates that either the profiles or the classification is inaccurate. In addition, around noon two days later (13/05), thick clouds were found but the aerosol retrieval does not show these clouds. This is surprising.

**P15, L21-22:** Trajectories ending at different altitudes were used, but how? You do not give information about that. Did you just average all maps for Figure 11?

**Section 5.2.1:** It would be interesting to see how theses maps change when the individual lifetimes are considered. Are CTM calculations of lifetimes at these days available?

**Fig 12:** Please add the data of the valdiating instruments, when possible.

**P17, L14-15:** How can I differ between the original data and the interpolated data in Fig 13? Please use other marker styles.

**P17, L16:** Highest values for southeast winds with southerly trajectories instead of north-westerly?

**P18, L18-19:** of on the order of hours --> in the order of hours

**Fig 14c:** The red dots are extremely small. Please increase the size of these dots.

**P31:** Wang, F et al.: Please change 2048 to the proper year.

---

## Referee Comment (RC2) · Anonymous Referee #2 · 10 Jan 2019

Wang et. al presented a MAX-DOAS observation for tropospheric vertical profiles of NO2, SO2, HONO, HCHO, CHOCHO and aerosols in the central-western North China Plain in May and June 2016. The MAX-DOAS results are validated comprehensively by the collocated measurements of ground based lidar, sun-photometer and in situ instrument, as well as overpass aircraft. Besides, characteristics of pollutants distribution and variations were analyzed combined with effects of regional and local transport.

As shown in the introduction, there were many studies of the trace gases and air pollutions of NCP in previous, also including the MAX-DOAS measurements. The main concerns is that what is the novelty or unique of this paper compared to the previous. I suggest the authors could highlight these in the manuscript.

Specific comments:
1. MAX-DOAS spectra analysis: It can be concluded from P5, Line27-28 that the authors used a spectrum measured in the zenith direction closest in time to the off-zenith measurements as a Fraunhofer reference spectrum. So if the telescope scanned in the sequence of 1°, 2°, 3°, 4°, 6°, 8°, 10°,15°, 20°,30°, 90°, the DSCDs of lower elevation angle (e.g. 1, 2, 3, 4) should use the zenith spectrum in previous scanning, but the DSCDs of higher elevation angle (e.g. 10, 15, 20, 30) use the zenith spectrum of current scanning. It means that the DSCDs of elevation angles in the same scanning were obtained with different reference spectrum. Any explanation or consideration about this treatment, which may bring some unknown effects in the profile retrieval procedure?

Fig.3: why the authors show the CHOCHO spectral analysis in another day compared with other species? And the CHOCHO absorption structure can not be well observed.

2. When you evaluated the DOAS data for HONO, did you consider the impurity of HONO in the NO2 reference spectra used? There is always some HONO in NO2 and that is subtracted in the DOAS algorithm. This leads to an underestimation of HONO by ca. 0.5% of the NO2, which can be significant during daytime and impacts the conclusions in your discussion about HONO/NO2.

3. Aerosol and trace gases retrieval:
   How was the vertical grids setting?
   How to distinguish the sky condition of high aerosols and clouds?
   In section 4.1, since the aerosol retrieval results were poor under the sky conditions of clear sky with high aerosols and cloudy sky (Fig. 6a and b), how to convince the trace gases retrieval are reliable?

All the reliable retrieval are the fundamental of the further analysis about effects of regional and local transport of pollutants.

Technical corrections:

P4, Line 28, "10:00 BT" change to "10:00 LT"

P5, Line 3-7, the results in Fig. 2 d were obtained from NASA website, however, the data in Fig. 2a, b and c? And the spatial resolution of the satellite products? Did the authors do any treatment or filter with the data? Please specify more clearly.

Fig. 2a, c, d, poor resolution. Please correct.

Fig. 7, I suggest the author present a panel plot of the differences of AE between MAX-DOAS and Lidar for more clearly and apparent comparison results.

Acknowledgements:

MAX-DOAS, LP-DOAS and etc. in Wuxi station? But the measurements was in NCP area.

WINDOAS software? But you used QDOAS

---

## Referee Comment (RC3) · Anonymous Referee #3 · 17 Jan 2019

The paper presents a comprehensive study of vertical distributions of NO2, SO2, HONO, HCHO, CHOCHO and aerosols by MAX-DOAS measurements during a spring/summer period (from 8 May to 10 June 2016) at a suburban site of the North China Plain. The profiles of these gases (volume mixing ratio) and aerosols (extinction coefficient) retrieved by MAX-DOAS are compared with the independent data, including in-situ measurements, Sun photometer, visibility meter, lidar and aircraft measurements. The effects of emissions and transport on the observed results are also

analyzed using the backward trajectories and various satellite data. The study is interesting, providing important information to the scientific community on air quality issue in eastern China. The paper is well written and organized, I would recommend the paper to be published subject minor revisions.

My major concern is on the comparison of the vertical profiles between ground-based MAX-DOAS and in situ aircraft measurements. While Sect. 5 devotes too much for a discussion about the regional and local transport of pollutants, more detailed analyses and discussions should have be added in Sect. 4.3 for the comparison of MAX-DOAS with aircraft measurements.

- Aerosol extinction and SO2 mixing ratio are underestimated significantly by MAX-DOAS with comparison to the aircraft measurements on 21 May 2016 (black dots in Fig. 8b). Why are the aircraft profiles, instead of MAX-DOAS profiles, converted (or "corrected") for better comparison? Since the airplane flew in a spiral route, were the chemical instruments stable enough to get reliable data with increasing air pressure? What is the vertical resolution (or precision) of the profile inversion by MAX-DOAS? The concept of the smoothing effect of the MAX-DOAS profile inversion should be discussed more in detail. I cannot find sufficient evidences in Sect. 4.2 to support the conclusion "The smoothing effect can cause MAX-DOAS retrievals to underestimate pollutants above 2 km and overestimate below" stated in Page 21, Line 31-32.

- It is stated that "the deviations between the MAX-DOAS and aircraft measurements can probably also be attributed to inhomogeneous horizontal distributions of pollutants and their temporal variation during a period of aircraft measurements" in Sect. 4.3 (Page 13, Line 14-16). Did you find any regular horizontal distribution patterns of aerosols and gases from aircraft measurements? Will the comparison improve if only the aircraft measurements in the area that the MAX-DOAS instrument was pointed to are selected?

- In addition to co-author's research group, other aircraft measurement work in the NCP
region should be credited, e,g, Ma et al. (2012) and Zhang et al. (2014); so did the MAX-DOAS measurement, e.g., Jin et al. (2016).

Technical issues: 1. Page 2, Line 32. What does "East-Aire" mean ?

2. Page 4, Line 23: What is the terrain height of the station?

3. Page 5, Line 15-20: The direction for the measurement should be mentioned.

4. Page 7, Line 12. Please check the punctuation here as well as elsewhere in the manuscript.

5. Page 8, Line 7-8. There are two references of Zhang et al., 2018. Please distinguish them when citing.

6. Page 9, Line 15-17. Please delete the repeating word of "be".

7. Page 12, Line 25-26. The agreement of the aerosol profiles from MAX-DOAS and lidar above 500m is not obvious, especially on 16 May, 2016. It is better to alter the color bar to show this point more clearly.

- 8. Page 12, Line 30. The content should move to the section 4.1.
- 9. Page 13, Line 20-22. This paragraph seems to be redundant.
- 10. Page 14, Line 4. The language expression needs to be improved.
- 11. Page 16, Line 6. Please note the subscript.

12. Page 16: Sect. 5.2.2. It is known that the MAX-DOAS measurements are performed during the daytime. However, the sorting here is mainly based on the nighttime trajectories. In addition, there are large differences between nighttime and daytime in Fig.S5, especially for the southerly trajectories.

- 13. Page 17, Line 34. Please change "Fig.12i" to "Fig.12l".
- 14. Page 22, Line 23, 26. Please check the names of the station "Wuxi" and software
"WINDOAS".

15. Figure 6: Please clarify the temporal resolution of the data used in Fig.6.

16. Table 2: "outliers"?

References

Jin, J., Ma, J., Lin, W., Zhao, H., Shaiganfar, R., Beirle, S., and Wagner, T.: MAX-DOAS measurements and satellite validation of tropospheric NO2 and SO2 vertical column densities at a rural site of North China, Atmospheric Environment, 133, 12-25, http://dx.doi.org/10.1016/j.atmosenv.2016.03.031, 2016.

Ma, J. Z., Wang, W., Chen, Y., Liu, H. J., Yan, P., Ding, G. A., Wang, M. L., Sun, J., and Lelieveld, J.: The IPAC-NC field campaign: a pollution and oxidization pool in the lower atmosphere over Huabei, China, Atmos. Chem. Phys., 12, 3883-3908, 10.5194/acp-12-3883-2012, 2012.

Zhang, W., Zhu, T., Yang, W., Bai, Z., Sun, Y. L., Xu, Y., Yin, B., and Zhao, X.: Airborne measurements of gas and particle pollutants during CAREBeijing-2008, Atmos. Chem. Phys., 14, 301-316, 10.5194/acp-14-301-2014, 2014.

---

## Author Comment (AC1) · 14 Mar 2019

**Reply to Ref. #1**

First of all we want to thank this reviewer for the positive assessment of our manuscript and the constructive and helpful suggestions.

General comments
Yang Wang et al. presented a comprehensive study of the temporal and spatial distribution of aerosols and trace gases in the central-western North China Plain. The manuscript is well structured and the results show good correlation with results of other instruments.
However, many results were filtered based on a cloud classification scheme which performs somehow unreliable to me. Especially low and high aerosol loads seem to be mixed up with clouds. To a certain points, MAX-DOAS profiles should be able to retrieve these differences but do not show larger deviations e.g. on 11/05 and 19/05. Since this classification has a large impact on the complete discussion, I would suggest to add a small estimate of the impact of wrongly classified/filtered scenarios on your results.
Furthermore, an additional analysis of NO2 retrieved in a different fitting window might help to clarify if horizontal inhomogeneties were present.

Author reply:
Many thanks for the positive assessment! We modified the manuscript based on the comments from you and the other two reviewers. The one-to-one replies are given in the following.
For your comments on cloud classification results, first of all, we agree on that some cases with clouds in the reality might be identified as "clear sky with high aerosols" or "low aerosols", due to certain thresholds are used. The problems can occur if the quantities which are used for the cloud identifications are close to these thresholds. Following your suggestion, we discussed the issue in the Sect. 2.2.4 of the revised manuscript as following:
"Since certain thresholds are used for the identification of cloud scenarios, two sky conditions might be interchanged because the derived quantities are close to the chosen thresholds. The problem occurs relatively often between the 'cloud free and high aerosol load' and 'continuous clouds' categories because they are only distinguished by the absolute value of the color index. The issue can impact the MAX-DOAS results of aerosol profiles and AODs due to the remaining cloud contamination. Fortunately the problem can be easily solved if an additional filter is applied, which is the convergence between measured and modelled $O_4$ dSCDs in the profile inversion for aerosols, based on the previous study in Wagner et al. (2016). If the convergence is bad, the corresponding aerosol results are possibly contaminated by clouds. Therefore the filter of convergence is applied to the MAX-DOAS results for the statistical analysis and elaborated in section 2.2.5 and Table 2.
In addition the issue can also impact the comparisons of MAX-DOAS results with coincident independent measurements under different sky conditions in section 4.1. However since the cases close to the thresholds do not dominate in each category, the general conclusions on the effects of clouds and aerosols are not significantly impacted."
Meanwhile we also modified the other part of Sect. 2.2.4 to illustrate the cloud classification scheme more clearly for the readers.
However for the mixing-up of the different sky condition on 11/05 and 19/05 which you pointed out, we think the phenomenon is not certainly due to the wrongly classified scenarios, but probably due to the true temporal variations of the sky conditions. You also pointed out that the

MAX-DOAS (aerosol) profiles do not show large deviations between the different sky conditions. This finding actually indicates the necessity of the cloud classification. The sky classification scheme is mainly based on the color index, which is ratio of intensities at 330nm against those at 390nm, while aerosol profile retrievals are based on $O_4$ absorptions. Our previous study indicates that usually the color index is more sensitive to sky conditions than the $O_4$ absorptions. This explains why the sky change of the condition changing is hardly seen from the aerosol profile results but is clearly seen from cloud classification results. Therefore the cloud-contaminated aerosol results need to be filtered out based on the cloud classification results.

Regarding horizontal inhomogeneous distributions of $NO_2$ and retrievals of $NO_2$ in the visible range, please see my reply to the specific comment 26.

**Specific Comments:**
1) P2, L14-19: Please add a reference to Fig 1. in the Introduction.
Author reply: we modified the manuscript by adding a reference to Fig 1 regarding the NCP region.
2) P2, L32: Please add the full name for the abbreviation East-Aire.
Author reply: The full name is East Asian Study of Tropospheric Aerosols: an International Experiment and added in the manuscript.

3) P3, L9: MAX-DOAS algorithms are not only based on OE. Iterative approaches like Newton-Gauß or Levenberg-Marquardt are in use. I would rather call these algorithms "inversion algorithms" or "inversion based algorithms".
Author reply: We modified the manuscript based on the suggestion as follows: "… inversion algorithms based on the optimal estimation (OE) method (and iterative approaches, e.g. the Newton-Gauß or Levenberg-Marquardt, are also used)"

4) P5, L17: Wang et al 2018 seems to be the wrong reference?
Author reply: Sorry for the mistake. The wrong reference is deleted in the revised manuscript.

5) P5, L26: "and for the MAD-CAT campagin" → "e.g. the MAD-CAT campaign"
Author reply: we modified the revised manuscript accordingly.

6) P5, L30: You state detection limits but not how they were calculated. Please add the missing information.
Author reply: Thanks for the suggestion! The detection limits are estimated based on the typical DOAS fit errors of individual species. The information is added in the revised manuscript.

7) Section 2.2.3: The uncertainties of the individiual parameters were given as percentages but where do they come from and how were they calculated? From a previous study or from not shown sensitivity tests? Please give some information.

Author reply: Thanks for pointing this out! The uncertainties of the profile retrievals due to a potential bias of the SSA and the Ångström exponent are derived based on sensitivity tests, which are not shown in the manuscript. We modified the manuscript to illustrate the information as the following:

"A systematic bias of the SSA typically contributes to an uncertainty of about 5% to the retrieved aerosol and trace gas profiles from MAX-DOAS measurements. These values were derived from sensitivity tests by varying SSA in the profile inversion."

"The uncertainty of the Ångström exponent (due to uncertainties of sun-photometer measurements) typically contributes to uncertainties of up to 20% to the retrievals of trace gas profiles. These results are derived from sensitivity tests by varying the Ångström exponent between 0.49 to 2.53 in the profile inversion. For the assumed range of the Ångström exponent see the discussion in section 3.1.1. "

8) P6, L11-12: When there is a sunphotometer measuring routinely. Why not using the exact SSA and asym parameter closest in time rather than averaged values? For which wavelengths are the averaged quantities? How did you convert to the proper wavelengths or did you assumed no wavelength dependency?

Author reply: The reason why we used the averaged values of SSA and the asymmetry parameter is due to the measurement uncertainties of the sun-photometer. Both parameters depend on the aerosol type, which is often similar for larger periods. Also there are many gaps in the measurement time series of the SSA and the asymmetry parameter due to the cloud filtering and quality controlling. In addition, both parameters are not measured in the UV spectral range, but retrieved at 440nm from the sun-photometer measurements. We add the missing information in the revised manuscript as follows:

"A fixed single scattering albedo (SSA) of 0.95 and an aerosol phase function parameterised according to Henyey and Greenstein (1941) with an asymmetry parameter of 0.72 are chosen according to averaged inversion results at 440nm from the sun-photometer also operated at the measurement station."

"It needs to be clarified that considering uncertainties of inversions of the SSA and asymmetry parameters of sun-photometer measurements, average values of both parameters are used in the inversion of MAX-DOAS measurements."

9) P6, L17: How were these wavelengths chosen? 354 for HONO is the mid of the fitting window but what about the other wavelengths? Was there are reason for not choosing the fitting window mid wavelengths?

Author reply: The wavelengths are the effective wavelengths of air mass factors of individual species in individual spectral ranges of the DOAS fits. The effective wavelengths can be calculated by weighting the wavelengths by the differential cross section values as shown in the previous study of Marquard et al. (2000) (Marquard, L. C., Wagner, T., & Platt, U. (2000). Improved air mass factor concepts for scattered radiation differential optical absorption spectroscopy of atmospheric species. Journal of Geophysical Research, 105(D1), 1315–1327. https://doi.org/10.1029/1999JD900340). We clarified this point in the revised manuscript as follows: "The air mass factors simulated by RTM are used for the profile inversion and the simulation wavelengths are calculated by weighting the wavelengths by the differential absorption cross section within the individual spectral ranges of the DOAS fits based on the method elaborated in the previous studies, e.g. Marquard et al. (2000)."

10) P6, L21: Why was the upper grid limit chosen to be at 3km? Typical altitudes from other studies are usually at 4km.

Author reply: Thanks for the asking! This is a mistake. In order to clarify the point, we modified the manuscript in the beginning of section 2.2.3 as the following:

"Tropospheric vertical profiles of aerosol extinction and volume mixing ratios (VMRs) of $NO_2$, $SO_2$, HONO, HCHO, and CHOCHO are retrieved from the elevation-dependent dSCDs by using the PriAM profile inversion algorithm (Wang et al., 2013a, b, 2017a) with a vertical grid of 200 m in an altitude range of up to 4 km. From the derived profiles the vertical column densities (VCD) of the trace gases and AODs are derived by vertical integrations. Due to the fact that no substantial information on the concentrations above 3km can be derived from the measurements, the retrieved profiles below 3 km are shown in all figures of the study."

11) P6, L25: Covariances of 100% of the surface value for all altitudes? Is this correct? The commonly used approach is a fixed percentage of the a priori profile of the individual altitudes.

Author reply: Thanks for pointing out this mistake! The manuscript is modified as:

"The diagonal elements of the a-priori covariances (Sa) at different altitudes are set as the square of 100% of the a-priori values at individual altitudes in order to balance the flexibility and stability of the profile inversion."

12) Table 2: Why is there a different SZA limit for SO2 compared to the other trace gases?

Author reply: The reason is that SO2 is retrieved at a shorter wavelength range compared to the other species. The intensity decreases stronger along increasing of SZA and the interference with the O3 absorption in the DOAS fit of SO2 is much stronger at a high SZA. We clarified this point in the revised manuscript as the following:

"Here it needs to be noted that a lower SZA threshold is set for the filtering of the $SO_2$ results than for the other species, because the intensity at short wavelengths is rather low and spectral interferences with the $O_3$ absorption increases strongly with SZA."

13) P6, L30: Here, you write R but in the Figure its R^2. Which one was given?

Author reply: Sorry for the mistake. The correlation parameter shown in the figures is R^2. In order to be consistent with other figures, we modified the figures and show R instead.

14) Fig 4: Please change the colors for either low or high aerosols in this and similar plots because it is hard to distinguish between both markers.
1. The aerosol retrieval shows similar profile shapes from 6 to 13 but the cloud classification finds different cloudy conditions, sometimes with thick clouds. How is it possible that the aerosol retrieval is not affected by thick clouds?
2. The a priori profiles for aerosols and SO2 are not even close to the the retrieved profiles. How can you be sure that you do not over- or underestimate the retrieved profiles due to in inaccurate a priori profile?
3. I do not understand why the degrees of freedom for aerosols are larger than any of the trace gas ds. This is unexpected for me. Could you please explain where these larger differences in ds for the invididual retrievals come from?

Author reply: the "cloud free with low aerosols" conditions do not appear on that day. Therefore we think it should not be a big problem to distinguish the blue and light blue colors on this day.

For your first comment, the maximum value of the color bar of 1 km⁻¹ is partly misleading. If the maximum value is enlarged to 2 km⁻¹, the effects of clouds can be clearly seen. The updated

figure is shown below. In addition, large extinctions at high altitudes can not be retrieved with certainty from MAX-DOAS measurements under optically thick and continuous clouds, because similar light paths can be expected at different elevation angles. In such cases, because the differential $O_4$ dSCDs at off-zenith views compared to the zenith view are used in the inversion of aerosol profiles, the differential $O_4$ dSCDs coming from the contributions of clouds are close to zero. Therefore no information on clouds can be derived from the differential $O_4$ dSCDs in such cases. However the identification of optically thick clouds is mainly based on the $O_4$ dSCDs in the zenith view compared to those under cloud-free sky conditions. Therefore the thick cloud conditions can be identified from MAX-DOAS.

[Figure]

For your second comment, we compared the AOD and near-surface values of $SO_2$ with the other co-located near-surface measurements. The comparisons verified the MAX-DOAS retrievals. We also show the comparisons of simulated and measured dSCDs in the supplement. If the profile inversion can well reproduce the dependences of dSCDs on elevation angle, the retrieved profiles are probably close to the truth. Further if MAX-DOAS measurements provide sufficient information on profiles, the retrieved profiles are expected to be different from the a-priori. The dependence of the profile inversion algorithm should on a-priori profile should be small. In the PriAM, we do the inversion in the logarithmic space and using the non-linear Levenberg-Marquardt iterative approach in order to reduce the constraints of a-priori.

For your third comment, since ds is the sum of the diagonal elements of the averaging kernel (A), and A can be calculated as the following equation:

$$A = \frac{\partial x}{\partial x_t} = GK = (S_a^{-1} + K^T S_\varepsilon^{-1} K)^{-1} K^T S_\varepsilon^{-1} K$$

For aerosol retrievals, K is the response of O4 dSCD to a variation of the aerosol extinction. Therefore A is a function of the aerosol extinction. For trace gas retrievals, K is the response of the trace gas dSCDs to the logarithmic concentration of the trace gas. Therefore A is a function of

the actual aerosols and the trace gas profiles, and A can be expected to be different for individual measurements and species. The reason why a larger ds is found for the aerosol retrieval than for the trace gases for the noon measurement shown in Fig. 4 is probably the fact that aerosols extend to higher altitudes than the trace gases. Actually a large ds can be also seen for the CHOCHO inversion (CHOCHO also extends to high altitudes).

15) P7, L15: Why is the near-surface extinction trusted under partially cloudy conditions but the AOD not? I would also assumed that the near-surface extinction is inaccurate when broken clouds led to a contamination of some elevation angles only. The profile will be smoothed due to the a priori smoothing and the retrieval for all altitudes should be affected.

Author reply: The filtering scheme is based on our previous study of long-term comparisons of MAX-DOAS aerosol results with sunphotometer and visibility meters in Wang et al. (2017a) (Wang, Y., Lampel, J., Xie, P., Beirle, S., Li, A., Wu, D., and Wagner, T.: Ground-based MAX-DOAS observations of tropospheric aerosols, NO2, SO2 and HCHO in Wuxi, China, from 2011 to 2014, Atmos. Chem. Phys., 17, 2189-2215, https://doi.org/10.5194/acp-17-2189-2017, 2017a.). The relevant figure in the paper is given below. It indicates that the systematic differences of the MAX-DOAS results of the near-surface aerosol extinction compared to the visibility-meter under all cloudy sky conditions are similar to those under cloud free sky conditions. We agree with you that broken clouds can impact near-surface aerosol results for individual measurements. Since the effects are different if clouds are observed at different elevation angles, and the impacted elevation angles are random, their contributions to obviously cancel out for the averages of long term measurements. In order to elaborate the point more clearly, we added the following information in the revised manuscript:

"Here it needs to be noted that clouds, especially broken clouds, can impact the MAX-DOAS results of the near-surface aerosol extinction for individual measurements. However since the cloud effects occur for different elevation angles, their overall impact is rather random. Therefore if long term measurements are averaged, cloud effects on the near-surface aerosol mostly cancel out and do not contribute to a systematic bias (Wang et al., 2017a)."

[Figure]

**Figure 9.** Mean absolute differences and standard deviations as well as correlation coefficients ($R$), slopes and intercepts derived from linear regressions of the AODs, and near-surface AE, $NO_2$ and $SO_2$ VMRs between MAX-DOAS and independent techniques for different sky conditions in autumn. The corresponding number of data points is shown in the bottom panel. Different colours denote AOD (compared with the Taihu AERONET level 1.5 data sets), AE (compared with the nearby visibility meter) and $NO_2$ and $SO_2$ (compared with the nearby long-path DOAS instrument). For calculations of mean absolute differences, the data derived from independent techniques are subtracted from those derived from the MAX-DOAS instrument. The data derived from the MAX-DOAS instrument are plotted against those derived from independent techniques for linear regressions.

16) P7, L25-28: The numbers are averages over the full time period? Please call it a total average then or specify what these numbers exactly mean.

Author reply: The numbers are averaged over the whole campaign period. Therefore we modified the sentence in the revised manuscript as suggested.

17) P8, L3-5: 5-times the word "also" in three lines. Maybe you can reformulate these two sentences.

Author reply: The sentences are modified in the revised manuscript.

18) P10, L6-7: I am confused of what you wrote in Section 2.2.3 and what you wrote in lines 6-7 (see comment P6, L11-12). Please explain your approach in greater detail.

Author reply: We modified the sentence in the revised manuscript as follows: "The Ångström parameter, single scattering albedo, and asymmetry factor are also retrieved from the sunphotometer measurements and are used as input for the inversion of the aerosol profiles from MAX-DOAS measurements, for details see section 2.2.3. "

In addition, we modified section 2.2.3 in the revised manuscript based on your comment P6, L11-12.

19) P10, L10: A parameter of 1 means an Angström Exponent of 1? Please replace the word "parameter". Furthermore, was the exponent of 1 used for all data or was the sunphotometer's Angström exponent closest in time applied for the conversion?

Author reply: We changed "parameter" to "Exponent". As mentioned above (your comment 8), considering the retrieval uncertainty of the sun-photometer measurements, the averaged value during the whole campaign period is used. We modified the sentence in the revised manuscript as follows:

"Aerosol extinction at 360 nm is derived from the visibility at 550 nm using an Ångström exponent of 1, which is the average value derived from the sun photometer measurements during the whole campaign period."

20) P11, L28-29: In addition to your reason, a general lower sensitivity for higher altitudes might be another reason. Furthermore, the limitation to 3km might also be an issue.

Author reply: We agree on that a general lower sensitivity for higher altitudes can also play a role here. Therefore we added the following sentence to the revised manuscript:

"Low sensitivity of MAX-DOAS measurements on aerosols located at high altitudes can also play a role. This finding is illustrated in Fig. 8b and discussed in section 4.3."

Concerning the effect of the limitation to 3km (see our reply to your comment 10), this was a wrong information. The inversion is done in the altitude range below 4km.

21) P12, L1-2: Generally, I would not assume that an elevated layer within the lowest kilometre can not be resolved by a MAX-DOAS profiling algorithm. For higher altitudes, this might be an issue. Is it possible to add a brief synthetic test on 3 or 4 elevated layers in different altitudes? Just to see the retrieval response. Because an elevated layer could also be possible when die underlying aerosol profile is box-like due to an oscillation around this box-features.

Author reply: We agree the explanation of the findings in the manuscript was not convincing. Based on this comment (and the comment 22), we modified the explanation as follows:

"The largest underestimation of the MAX-DOAS results compared to the visibility-meter results is found in the morning and when aerosol loads are large. Since the boundary layer height is lower in the morning, larger vertical gradients of aerosols in the layer between 0 and 200m can be expected. The different air mass measured by the MAX-DOAS and the visibility-meter and a large vertical gradient near the surface might be the reason for the large underestimation of the MAX-DOAS results compared to the visibility-meter results."

22) Fig 6:

1. It would be interesting to know if these outliers for high visibilitymeter AE correspond to certain geometries, time or weather conditions?

2. P12, L11: When this behaviour for different cloudy conditions can be attributed to clouds, why is there a different correlation for NO2 and SO2 for high aerosols and cloudy sky? NO2 has a better correlation for cloudy sky while SO2 has a better correlation for high aerosols.

Do you expect large inaccuracies in the classification scheme for clouds and aerosols?

Author reply: For your comment 1, many thanks for the good suggestion to see the dependence on the time of the day! We modified Fig. 6 in the revised manuscript by adding colors to show the time of the day. For the comparison of the near-surface AEs, we can see that the largest underestimation of MAX-DOAS results compared to visibility meter results occurs in the morning and when the aerosol load is large. Since the boundary layer height is lower in the morning, larger vertical gradient of aerosols can be expected. In the lowest vertical layer between 0 and 200 m, since aerosols might accumulate near the surface. The visibility meter measures aerosols near the surface, but the MAX-DOAS inversion results represent averages of the

aerosols extinction in each layer. The different air masses measured by MAX-DOAS and the visibility-meter together with large vertical gradients in the lowest layer between 0 and 200 m might be the reason for the underestimation of the MAX-DOAS results compared to the visibility-meter results. The revised manuscript is modified as the following:

"The comparison of the near-surface aerosol extinction between the MAX-DOAS measurements and the visibility-meter is shown in Fig 6b. While very good agreement is found for the category "clear sky with low aerosols" (R of 0.81 and slope of 1.02), much worse correlation is found for "clear sky with high aerosols" (R of 0.32 and slope of 0.14). Even worse agreement is found for the category "cloudy sky" (R of 0.22 and slope of 0.11). The largest underestimation of the MAX-DOAS results compared to the visibility-meter results is found in the morning and when aerosol loads are large. Since the boundary layer height is lower in the morning, larger vertical gradients of aerosols in the layer between 0 and 200m can be expected. The different air mass measured by the MAX-DOAS and the visibility-meter together with strong vertical gradients near the surface might be the reason for the large underestimation of the MAX-DOAS results compared to the visibility-meter results. In addition the effect of clouds on the MAX-DOAS aerosol retrievals are probably the reason for the larger scattering under the cloudy sky conditions compared to cloud-free sky conditions. Note that clouds are not included in the RTM which is used for profile retrievals of aerosols and trace gases."

For your comment 2, the different cloud effects for NO2 and SO2 can be attributed to their different vertical distributions. As shown in Fig. 12, SO2 typically extends to higher altitudes than NO2. Therefore cloud effects, especially for broken clouds, on the near-surface SO2 concentration can be expected to be stronger than for NO2. Moreover, for NO2, the photolysis rate can be expected to be lower under cloudy conditions than under cloud-free conditions. Therefore the vertical distributions of NO2 in the layer between 0 and 200 m might be smoother under cloudy conditions than under cloud-free conditions. The smoother distribution might be the reason for the better agreement of the NO2 results between MAX-DOAS and in-situ measurements. Although clouds can also affect the NO2 profile inversion, since NO2 is close to the surface, the effect of a smoother vertical distribution of NO2 under cloudy conditions might be important for the comparison of MAX-DOAS and in-situ results of NO2. In order to clarify this point, we added the following text in the revised manuscript:

"Further, different cloud effects on NO2 and SO2 comparisons are found. For $SO_2$ worse correlation is found under "cloudy sky" conditions than under "clear sky with high aerosols", while a better correlation is found for $NO_2$. Effects of clouds in the profile retrievals of SO2 can be expected to be stronger than for NO2 due to the fact that $SO_2$ can extend to higher altitudes than $NO_2$ as shown in Fig. 12. In addition, the photolysis rate of $NO_2$ can be expected to be lower under cloudy conditions than under cloud-free sky conditions. Therefore the vertical distributions of $NO_2$ in the layer between 0 and 200 m might be smoother under cloudy conditions than under cloud-free sky condition. The smoother vertical distribution is probably the reason for the better agreement of the $NO_2$ results between the MAX-DOAS results and in-situ measurements."

23) P12, L12-14: A limited vertical sensitivity for near-surface trace gas concentrations might also be important when the trace gas is concentrated in a shallow layer much smaller than the grid step width of the profiling algorithm, especially when using a coarse grid steps width of 200m.

Author reply: We agree on the suggestion. We want to say the same thing with the sentence of "This finding is probably related to vertical and horizontal inhomogeneity of the species". In

order to elaborate it more clearly, we added a further explanation in the bracket as "(especially in the lowest vertical layer between 0 and 200 m)" in the revised manuscript.

24) Section 4.2:
    1. What is the vertical resolution of the Lidar?
    2. Was AK smoothing applied for the Lidar measurements (with the assumption that the vertical resolution is higher than that for MAX-DOAS)? Please add also the Lidar AOD to the AOD sub-figure.
    3. Why are high near-surface values not similarly found by both instruments? The sensitivity for near-surface values should be the highest for both instruments and different air masses and clouds are not expected to be that important for lower altitudes. E.g. MAX-DOAS found larger extinctions at 15:00 (2016/5/16) while the Lidar found larger values in the evening of 2016/5/17.

Author reply: 1) The vertical resolution of the Lidar measurements is 7.5m. The information is added in the revised manuscript.

2) The Lidar results shown in Fig. 7 are the original results without the smoothing by the AK of the MAX-DOAS measurements. Since you asked for it, we plot the Lidar profile smoothed by an AK in the bottom subfigure of the plots below. We also show the lidar profiles interpolated to the grid of MAX-DOAS profile retrievals in the fifth subfigure of the plots below. Since the plots of smoothed profiles do not give more important information compared to the original profiles in the comparisons with MAX-DOAS results, we decide not to show the smoothed profiles.

The AODs (green) calculated from the original Lidar aerosol profiles are also given in the plots below. Since there is a blind area of the lidar measurements below 500m, the AODs derived from the Lidar profiles are not suitable for the quantitative comparison with the MAX-DOAS AOD results due to large amount of aerosols located at altitudes below 500m. Therefore we prefer not to show the Lidar AODs in Fig. 7 of the manuscript.

[Figure]

[Figure]

3) First of all, different air masses are measured by the MAX-DOAS and Lidar instruments. MAX-DOAS pointed to the North with a typical horizontal effective light path of about 5 to 10 km. In contrast, the Lidar pointed to the zenith. Horizontal inhomogeneous distributions of aerosols might play a role on the differences between the two techniques. Secondly, as you mentioned in your previous comment, broken clouds can impact some elevation angles of MAX-DOAS measurements. The same broken clouds are probably not be seen by the Lidar in the zenith. The effects of inhomogeneous cloud coverage can also contribute to the differences between the two techniques. The potential reasons for the differences have been given in the manuscript as follows:

"The remaining differences can probably be explained by the fact that different air masses are observed by both techniques, while horizontal inhomogeneities of aerosols and cloud cover could appear. For example, different clouds and aerosols could be observed by both instruments. Note that the MAX-DOAS telescope was pointed towards the North, while the Lidar measured the atmosphere directly above the station. The sun-photometer measured the air masses in the direction of the sun."

25) P13, L8: How did you calculate these "combined profiles"? Linear interpolation between lowest air-craft and surface value? Please add some information.

Author reply: Your understanding is correct. A linear interpolation is used. We added the information in the revised manuscript as follows:

"Combined profiles" are generated by combining averaged aircraft profiles with averaged surface measurements. Values at altitudes between the lowest aircraft measurements and surface measurements are generated by linear interpolations."

26) Fig 8:

1. Please add times for MAX-DOAS measurements and the overpasses. It would also be nice to have some color-coding or different grey-scales for the green surface values to identify the surface value changes throughout the measurement period.

2. Why do the SO2 curves always agree better than NO2 even though the degrees of freedom are much lower for SO2? Did you try to retrieve NO2 also in another fitting window (> 400nm)? It would be interesting to see if the results differ strongly. That would support the argument of horizontal inhomogeneities.

Author reply: 1) The times of the comparisons are added in the revised manuscript and figure. Regarding color-coding of the in-situ data, we prefer not to follow this suggestion, because the current plots are already quite busy. The motivation to show the in-situ data over the comparison period is to illustrate the temporal variability of the pollutants near the surface. Thus the time information might be not important here. In addition if the readers want to see the time series of the in-situ measurements, they can find them in Fig. S3 of the supplement.

2) An important reason why the aircraft NO2 results are much larger than the MAX-DOAS results might be the interference of NOy with NOx in the aircraft measurements. The same effect can be seen from the comparison with the surface in-situ measurements of NO2 shown in Fig. 6c. The same technology is applied to the aircraft and ground based measurements. We added this information in the revised manuscript as follows:

"For $NO_2$, significantly larger values of the aircraft measurements than the MAX-DOAS measurements can be seen on 21 May. Since the same technique as for the ground based in-situ measurements of NO2 (see section 4.1) was used in the aircraft measurements, the interference of NOy with NOx might also cause an overestimation of the $NO_2$ concentrations derived from the aircraft measurements."

We tried to retrieve NO2 in the visible range, the general time series are consistent between the UV and visible range. I think it needs a sophisticated inversion algorithm to retrieve horizontal inhomogeneity from $NO_2$ absorptions in the UV and Visible. This is beyond the scope of the study.

27) P13, L24-25: Please add the time when the photos were taken on the individual days.
Author reply: the photos were taken around noon on all the days. The information is given in Fig. 9.

28) Section 5.1: Why does the cloud classification show highly variable results between 6 and 14 BT (11/05) but the aerosol profiles do not differ strongly? This indicates that either the profiles or the classification is inaccurate. In addition, around noon two days later (13/05), thick clouds were found but the aerosol retrieval does not show these clouds. This is surprising.

Author reply: For the question regarding 11/05, the answer has been given in the reply to comment 14. The same reason can explain the findings on 13 May. If the continuous clouds are located at high altitudes, similar absorption paths of $O_4$ in the cloud for the zenith view and off-zenith view of MAX-DOAS measurements can be expected. Therefore with the zenith spectrum as the FRS in the $O_4$ DOAS fits of off-zenith spectra, the partial $O_4$ dSCDs due to the contribution of the cloud will cancel out. Therefore no clouds can be retrieved under such conditions using the $O_4$ dSCDs at off-zenith views. We show the backscattering signal measured by a collocated ceilometers in the figure below. It indicates that the bottom height of clouds on 13 May is around 8km. By comparing the cloud classification results with the ceilometers results, we can conclude that the cloud classification results are correct.

[Figure]

We present the comparisons of measured and modelled O4 dSCDs derived from aerosol profile retrievals of MAX-DOAS in the bottom of the figure below. We can see good agreement of the measured and modelled $O_4$ dSCD. Therefore we conclude that the results of the profile inversion should be realistic.

[Figure]

29) P15, L21-22: Trajectories ending at different altitudes were used, but how? You do not give information about that. Did you just average all maps for Figure 11?
Author reply: Your understanding is correct. We generated the maps with trajectories ending at different altitudes. Then the maps were averaged to generate Fig. 11. In order to clarify this procedure, we added the following text in the revised manuscript:
"In order to consider pollution transport at different altitudes, we generated individual maps using trajectories ending at 100m, 300m, 500m, 1km, and 2km above the measurement site. These maps are then averaged to generate the final map. The individual maps using trajectories ending at different altitudes are presented in Fig. S4 in the supplement. The final maps are shown in Fig. 11."

30) Section 5.2.1: It would be interesting to see how theses maps change when the individual lifetimes are considered. Are CTM calculations of lifetimes at these days available?
Author reply: We showed the maps with different backward times in Fig. S4 of the supplement. We use different backward times for the generation of the maps to implicitly test the effects of varying lifetime. Note that no information about the lifetime is available from the CTM in this study. In addition, we don't think that the method could be in general be improved if information on the lifetime would be used to scale the trace gas VCDs for the generations of the maps, because the trace gas columns measured by MAX-DOAS are actually transported from areas in

different distances. It is unknown which fraction of the total trace gas amount was transported from an area in a specific distance. Therefore the scaling of the VCD based on the lifetime can even cause artificially high VCDs in the generated maps if large backward time is used.

31) Fig 12: Please add the data of the validating instruments, when possible.
Author reply: Here we did not follow the reviewer's suggestion, because we feel that the plots would be too busy, while at the same time no relevant information will be added. Here it should be noted that the comparisons with independent measurements have been already shown in Fig. 6. Moreover, the study of the effects of transports is only based on the MAX-DOAS results.

32) P17, L14-15: How can I differ between the original data and the interpolated data in Fig 13? Please use other marker styles.
Author reply: Sorry for the missing information in the manuscript. The data shown in Fig. 13 are all original data. No interpolated data are shown. Therefore we delete the sentence "Note that the data are interpolated to provide a more consistent overview. The original data are also plotted in Fig. 13 in order to show the representativeness of the interpolated data." in the revised manuscript.

33) P17, L16: Highest values for southeast winds with southerly trajectories instead of north-westerly?
Author reply: The higher values are found for southeast winds when the long-range trajectories are from the northwest. This finding indicates the effect of short-range transport of pollutants emitted in the downtown area of Xingtai (with an iron factory) about 20km southeast of the station. The effect of short-range transport can only be seen when the air mass over the station is dominated by transport of clean air masses from the northwest. In order to describe it more clearly, we modified the sentence in the revised manuscript as follows:
"In general the higher values of aerosols, $NO_2$, $SO_2$, and HCHO occur for the southeast wind directions than for other wind directions on the days of north-westerly trajectories. This finding possibly indicates the effect of short-range transports of pollutants emitted in the downtown area of Xingtai (with an iron factory) about 20km southeast of the station. The effect of short-range transport can be well identified when the air mass over the station is dominated by transport of clean air mass from the northwest."

34) P18, L18-19: of on the order of hours --> in the order of hours
Author reply: It is corrected in the revised manuscript.

35) Fig 14c: The red dots are extremely small. Please increase the size of these dots.
Author reply: The images with the red dots are downloaded from https://worldview.earthdata.nasa.gov. Therefore we can't change the size. We added the website for Fig. 14c in the revised manuscript.

36) P31: Wang, F et al.: Please change 2048 to the proper year.
Author reply: It is modified as "2018" in the revised manuscript.

---

## Author Comment (AC2) · 14 Mar 2019

**Reply to Ref. #2**

First of all we want to thank this reviewer for the positive assessment of our manuscript and the constructive and helpful suggestions.

General comments
Wang et. al presented a MAX-DOAS observation for tropospheric vertical profiles of NO2, SO2, HONO, HCHO, CHOCHO and aerosols in the central- western North China Plain in May and June 2016. The MAX-DOAS results are validated comprehensively by the collocated measurements of ground based lidar, sun-photometer and in situ instrument, as well as overpass aircraft. Besides, characteristics of pollutants distribution and variations were analyzed combined with effects of regional and local transport. As shown in the introduction, there were many studies of the trace gases and air pollutions of NCP in previous, also including the MAX-DOAS measurements. The main concerns is that what is the novelty or unique of this paper compared to the previous. I suggest the authors could highlight these in the manuscript.

Author reply:
Many thanks for the suggestion! We modified the manuscript based on the comments from you and the other two reviewers. The one-to-one replies are given in the following part. For your main concern, we followed your suggestion to highlight the novelty and unique point of the study in the abstract and introduction as follows:
"Note that although several MAX-DOAS measurements of trace gases and aerosols in the NCP area have been reported in previous studies, this study is the first work to derive a comprehensive set of vertical profiles of $NO_2$, $SO_2$, HONO, HCHO, CHOCHO, and aerosols from measurements of one MAX-DOAS instrument. Also, so far the validation of MAX-DOAS profile results by comparison with various surface in-situ measurements as well as profile measurements from Lidar and aircraft is scarce. Moreover, the backward propagation approach to characterize the contributions of regional transport of pollutants from different regions was for the first time applied to the MAX-DOAS results of trace gases and aerosols."

**Specific Comments:**
1) MAX-DOAS spectra analysis: It can be concluded from P5, Line27-28 that the authors used a spectrum measured in the zenith direction closest in time to the off-zenith measurements as a Fraunhofer reference spectrum. So if the telescope scanned in the sequence of 1°, 2°, 3°, 4°, 6°, 8°, 10°,15°, 20°,30°, 90°, the DSCDs of lower elevation angle (e.g. 1, 2, 3, 4) should use the zenith spectrum in previous scanning, but the DSCDs of higher elevation angle (e.g. 10, 15, 20, 30) use the zenith spectrum of current scanning. It means that the DSCDs of elevation angles in the same scanning were obtained with different reference spectrum. Any explanation or consideration about this treatment, which may bring some unknown effects in the profile retrieval procedure? Fig.3: why the authors show the CHOCHO spectral analysis in another day compared with other species? And the CHOCHO absorption structure can not be well observed.
Author reply: Regarding the Fraunhofer reference spectrum, thanks for pointing out the obscure elaboration! We modified the description in the revised manuscript as follows: "A sequential Fraunhofer reference spectrum, which is derived from interpolation of two zenith spectra measured before and after an elevation sequence to the measurement time of individual off-zenith measurements, is used in the DOAS fits.". Regarding the HCHO spectral fit shown in Fig. 3, the CHOCHO dSCD around noon on 27 May is the highest during the whole campaign. As you have seen, CHOCHO fit is quite difficult to analyse, its largest optical depth is only ~0.001, two orders smaller than the optical depth of $NO_2$. In order to show the best fit, we showed the results on 27 May in Fig. 3. We clarified the point in the revised manuscript for Fig. 3 as follows: "Note that the CHOCHO fit shown in the figure is for the largest CHOCHO dSCD retrieved around noon during the whole campaign period."

2) When you evaluated the DOAS data for HONO, did you consider the impurity of HONO in the NO2 reference spectra used? There is always some HONO in NO2 and that is subtracted in the DOAS algorithm. This leads to an underestimation of HONO by ca. 0.5% of the NO2, which can be significant during daytime and impacts the conclusions in your discussion about HONO/NO2.

Author reply: We searched the literature regarding measurements of $NO_2$ cross sections, namely $NO_2$ reference spectra. However we found no publications reporting the effects of the contamination of HONO in the $NO_2$ cell. In addition if there are HONO structures in the $NO_2$ cross section, we can expect an increase of (negative) HONO dSCDs along the increase of $NO_2$ dSCDs during the day. However we don't see such an increase. Therefore we think the HONO impurity effect on the calculations of the HONO/$NO_2$ ratio in the study is negligible. If the reviewer knows a publication about the HONO impurity issue, please inform us.

3) Aerosol and trace gases retrieval:
How was the vertical grids setting?
How to distinguish the sky condition of high aerosols and clouds?
In section 4.1, since the aerosol retrieval results were poor under the sky conditions of clear sky with high aerosols and cloudy sky (Fig. 6a and b), how to convince the trace gases retrieval are reliable? All the reliable retrieval are the fundamental of the further analysis about effects of regional and local transport of pollutants.

Author reply: The vertical grid is 200 m. The information is added in the revised manuscript.
Regarding the cloud classification, the difference between "clear sky with high aerosol load" and "continuous clouds" is the spread of the color index at different elevation angles of the MAX-DOAS measurements. The spread is much smaller under "continuous clouds" than under "clear sky with high aerosol load". The difference between "clear sky with high aerosol load" and "broken clouds" is the temporal variations of the color index measured by MAX-DOAS. Because the cloud coverage can change rapidly under "broken clouds", the temporal variation is much larger under "broken clouds" than under "clear sky with high aerosol load". We elaborated the details of the method in our previous publications of "Wagner, T., Beirle, S., Dörner, S., Friess, U., Remmers, J. and Shaiganfar, R.: Cloud detection and classification based on MAX-DOAS observations, Atmos. Meas. Tech., 7, 1289-1320, 2014" and "Wagner, T., Beirle, S., Remmers, J., Shaiganfar, R., and Wang, Y.: Absolute calibration of the colour index and O4 absorption derived from Multi AXis (MAX-)DOAS measurements and their application to a standardised cloud classification algorithm, Atmos. Meas. Tech., 9, 4803-4823, https://doi.org/10.5194/amt-9-4803-2016, 2016.".
Regarding the cloud effect, since clouds typically located at altitudes above the trace gases, clouds have usually a stronger impact on the $O_4$ absorptions than on the trace gases. Therefore they impact the aerosol retrievals stronger than the trace gas retrievals.
Under high aerosol load conditions, the discrepancy between the aerosol results from MAX-DOAS and sun-photometer and visibilitymeter measurements are probably mainly due to inhomogeneous horizontal distributions and different air masses measured by the different instruments. In addition, MAX-DOAS might underestimate aerosols at high altitudes due to the low sensitivity of MAX-DOAS measurements there. Since trace gases typically located at low altitudes, it is not probable that the underestimation of aerosols at high altitudes by MAX-DOAS significantly impacts the trace gas profile retrievals.

Technical corrections:
1) P4, Line 28, "10:00 BT" change to "10:00 LT"
Author reply: Thanks for pointing it out. BT is Beijing time. In order to clarify the point, we added a sentence in the revised manuscript as follows: "Since the longitude difference of the station and Beijing is only 2°, the Beijing time is almost the local time.".

2) P5, Line 3-7, the results in Fig. 2 d were obtained from NASA website, however, the data in Fig. 2a, b and c? And the spatial resolution of the satellite products? Did the authors do any treatment or filter with the data? Please specify more clearly.

Author reply: Thanks for pointing out the missing information. We modified the paragraph regarding the satellite data in section 2.1 to add the information in the revised manuscript. The modified paragraph is the following:

"Averaged maps of $NO_2$ (from DOMINO v2, Boersma et al., 2007 and 2011), $SO_2$ (from BIRA-IASB, Theys et al., 2015), and HCHO (from BIRA-IASB, De Smedt et al., 2008, 2012 and 2015) derived from satellite observations of the Ozone Monitoring instrument (OMI) (Levelt et al., 2006a and b) for May and June during the period 2012 to 2016 for the same area as shown in Fig. 1a are shown in Fig. 2a, b, and c, respectively. The spatial resolution of the OMI data is $13 \times 24$ $km^2$ in nadir. Note that the OMI data of the outermost pixels (i.e. pixel numbers 1–5 and 56–60) and pixels affected by the so-called "row anomaly" (see http://www.temis.nl/airpollution/no2col/warning.html) were removed. In Fig. 2d a map of the averaged aerosol optical depths (AODs) at 550 nm derived from the Moderate Resolution Imaging Spectroradiometer (MODIS) (Kaufman et al., 2002) for the same period is shown (provided by NASA on http://ladsweb.nascom.nasa.gov/data/search.html). The spatial resolution of the MODIS AOD data is $5 \times 5$ $km^2$. In order to exclude cloud contaminated data, for both OMI and MODIS data, only the data with cloud fractions smaller than 30% are included for the generation of the maps. A grid interval of 0.02 °is used to generate the averaged maps of the OMI and MODIS data by binning the satellite data of pixels around each grid with distance weightings."

3) Fig. 2a, c, d, poor resolution. Please correct.

Author reply: Since the pixels of OMI satellite instruments cover an area of $13 \times 24 km^2$, the map resolution can not be further improved.

4) Fig. 7, I suggest the author present a panel plot of the differences of AE between MAX-DOAS and Lidar for more clearly and apparent comparison results.

Author reply: We followed the idea and added the panel in the new Fig. 7 in the revised manuscript.

5) Acknowledgements:
   MAX-DOAS, LP-DOAS and etc. in Wuxi station? But the measurements was in
   NCP area.
   WINDOAS software? But you used QDOAS

Author reply: Thanks for pointing out the mistakes! The mistakes are corrected in the revised manuscript.

---

## Author Comment (AC3) · 14 Mar 2019

**Reply to Ref. #3**

First of all we want to thank this reviewer for the positive assessment of our manuscript and the constructive and helpful suggestions.

General comments
The paper presents a comprehensive study of vertical distributions of NO2, SO2, HONO, HCHO, CHOCHO and aerosols by MAX-DOAS measurements during a spring/summer period (from 8 May to 10 June 2016) at a suburban site of the North China Plain. The profiles of these gases (volume mixing ratio) and aerosols (extinction coefficient) retrieved by MAX-DOAS are compared with the independent data, including in-situ measurements, Sun photometer, visibility meter, lidar and aircraft measurements. The effects of emissions and transport on the observed results are also analyzed using the backward trajectories and various satellite data. The study is interesting, providing important information to the scientific community on air quality issue in eastern China. The paper is well written and organized, I would recommend the paper to be published subject minor revisions. My major concern is on the comparison of the vertical profiles between ground-based MAX-DOAS and in situ aircraft measurements. While Sect. 5 devotes too much for a discussion about the regional and local transport of pollutants, more detailed analyses and discussions should have be added in Sect. 4.3 for the comparison of MAX-DOAS with aircraft measurements.

Author reply:
Many thanks for the positive assessment! We modified the paper based on the comments from you and the other two reviewers. Please see the replies and modifications regarding your specific comments below.

**Specific Comments:**
1) - Aerosol extinction and SO2 mixing ratio are underestimated significantly by MAX-DOAS with comparison to the aircraft measurements on 21 May 2016 (black dots in Fig. 8b). Why are the aircraft profiles, instead of MAX-DOAS profiles, converted (or "corrected") for better comparison? Since the airplane flew in a spiral route, were the chemical instruments stable enough to get reliable data with increasing air pressure? What is the vertical resolution (or precision) of the profile inversion by MAX-DOAS? The concept of the smoothing effect of the MAX-DOAS profile inversion should be discussed more in detail. I cannot find sufficient evidences in Sect. 4.2 to support the conclusion "The smoothing effect can cause MAX-DOAS retrievals to underestimate pollutants above 2 km and overestimate below" stated in Page 21, Line 31-32.
Author reply: Thanks for the comment! We give the answers to your individual questions below:
Question 1: "Why are the aircraft profiles, instead of MAX-DOAS profiles, converted (or "corrected") for better comparison?"
Answer: The question might be related to the unclear explanation in the manuscript. We modified the sentence in the revised manuscript as follows:
"Since the limited response of MAX-DOAS profile retrievals to the true profiles, the retrieved profile xˆ can be represented as the true profile x, smoothed by the AK according to the equation: xˆ = xa + AK(x − xa), where xa is a-priori profile used in the profile retrieval of MAX-DOAS. To account for the smoothing effect of the MAX-DOAS profile inversion in the comparisons, the AKs of the MAX-DOAS profile retrievals are applied to the averaged aircraft profiles, which are treated as the true atmospheric profile x to generate the "smoothed profiles" xˆ. Additionally the combined profiles, derived from the averaged aircraft profile and surface data, are considered as the true atmospheric profile x and converted to "smoothed combined profiles" using the AK of the MAX-DOAS profile retrievals. The "smoothed profiles" and "smoothed combined profiles" are shown in Fig. 8. By comparing the smoothed profiles with the original profiles derived from the aircraft measurements, the smoothing effect of MAX-DOAS retrievals can be evaluated."

Question 2: Since the airplane flew in a spiral route, were the chemical instruments stable enough to get reliable data with increasing air pressure?

Answer: The $NO_2$ analyzer have internal pressure controllers that maintain the pressure constant at 128 torr, well below the pressure altitudes we flew. So their measurements are not affected by the ambient pressure changes at all. All other trace gas analyzers like ozone, $SO_2$, NO, and NOy are corrected for pressure and temperature when they reported the final concentrations. So our instruments are stable enough to make reliable measurements during the spiral profiles. All the aircraft instruments have been used for airborne measurements in the United States and China (e.g. Taubman et al., 2006; Dickerson et al., 2007; Hains et al., 2008; He et al., 2012; He et al., 2014; Ren et al., 2018; Salmon et al., 2018).

Taubman, B. F., Hains, J. C., Thompson, A. M., Marufu, L. T., Doddridge, B. G., Stehr, J. W., Piety, C. A., and Dickerson, R. R.: Aircraft vertical profiles of trace gas and aerosol pollution over the mid-Atlantic United States: Statistics and meteorological cluster analysis, Journal of Geophysical Research-Atmospheres, 111, D10s07 10.1029/2005jd006196, 2006.

Hains, J. C., Taubman, B. F., Thompson, A. M., Stehr, J. W., Marufu, L. T., Doddridge, B. G., and Dickerson, R. R.: Origins of chemical pollution derived from Mid-Atlantic aircraft profiles using a clustering technique, Atmospheric Environment, 42, 1727-1741, 10.1016/j.atmosenv.2007.11.052, 2008.

He, H., Li, C., Loughner, C. P., Li, Z., Krotkov, N. A., Yang, K., Wang, L., Zheng, Y., Bao, X., Zhao, G., and Dickerson, R. R.: SO2 over central China: Measurements, numerical simulations and the tropospheric sulfur budget, Journal of Geophysical Research: Atmospheres, 117, doi:10.1029/2011JD016473, 2012.

He, H., Loughner, C. P., Stehr, J. W., Arkinson, H. L., Brent, L. C., Follette-Cook, M. B., Tzortziou, M. A., Pickering, K. E., Thompson, A. M., Martins, D. K., Diskin, G. S., Anderson, B. E., Crawford, J. H., Weinheimer, A. J., Lee, P., Hains, J. C., and Dickerson, R. R.: An elevated reservoir of air pollutants over the Mid-Atlantic States during the 2011 DISCOVER-AQ campaign: Airborne measurements and numerical simulations, Atmospheric Environment, 85, 18-30, 10.1016/j.atmosenv.2013.11.039, 2014.

Ren, X., Salmon, O. E., Hansford, J. R., Ahn, D., Hall, D., Benish, S. E., Stratton, P. R., He, H., Sahu, S., Grimes, C., Heimburger, A. M. F., Martin, C. R., Cohen, M. D., Stunder, B., Salawitch, R. J., Ehrman, S. H., Shepson, P. B., and Dickerson, R. R.: Methane Emissions From the Baltimore-Washington Area Based on Airborne Observations: Comparison to Emissions Inventories, Journal of Geophysical Research: Atmospheres, 0, doi:10.1029/2018JD028851, 2018.

Salmon, O. E., Shepson, P. B., Ren, X., He, H., Hall, D. L., Dickerson, R. R., Stirm, B. H., Brown, S. S., Fibiger, D. L., McDuffie, E. E., Campos, T. L., Gurney, K. R., and Thornton, J. A.: Top-Down Estimates of NOx and CO Emissions From Washington, D.C.-Baltimore During the WINTER Campaign, Journal of Geophysical Research: Atmospheres, 123, 7705-7724, doi:10.1029/2018JD028539, 2018.

Question 3: What is the vertical resolution (or precision) of the profile inversion by MAX-DOAS?

Answer: the vertical resolution can be represented by the averaging kernels. The resolution is shown in the last paragraph of section 2.2.3 and Fig. 4c. In order to introduce the meaning of the averaging kernel more clearly, we modified the sentence in the revised manuscript as follows:

"The vertical resolution and sensitivities of the retrievals at different altitutdes can be quantified by the so-called averaging kernel matrix $AK = \partial x\hat{} /\partial x$, which represents the sensitivity of the retrieved profile x^ as a function of the true atmospheric profile x. The typical AK of the profile inversions shown in Fig. 4c indicate that the sensitivity of the profile retrievals of trace gases and aerosols systematically decreases with altitude. "

Question 4: The concept of the smoothing effect of the MAX-DOAS profile inversion should be discussed more in detail. I cannot find sufficient evidences in Sect. 4.2 to support the conclusion "The smoothing effect can cause MAX-DOAS retrievals to underestimate pollutants above 2 km and overestimate below" stated in Page 21, Line 31-32.

Answer: In order to discuss the smoothing effect better, we added a sentence in Section 4.3 in the revised manuscript as follows: "Generally, pollutants above 1km are significantly underestimated due to the

smoothing effect of MAX-DOAS profile retrievals." And the sentence in the conclusion section is modified as "The smoothing effect of MAX-DOAS profile retrievals can cause a reshaping of box-profiles below 2km towards exponentially decreasing profiles. This effect can cause MAX-DOAS measurements significantly underestimate pollutants located at altitudes above 1km.".

2) - It is stated that "the deviations between the MAX-DOAS and aircraft measurements can probably also be attributed to inhomogeneous horizontal distributions of pollutants and their temporal variation during a period of aircraft measurements" in Sect. 4.3 (Page 13, Line 14-16). Did you find any regular horizontal distribution patterns of aerosols and gases from aircraft measurements? Will the comparison improve if only the aircraft measurements in the area that the MAX-DOAS instrument was pointed to are selected?

Author reply: A large variability of the original data from the aircraft measurements at individual altitudes can be seen in Fig. 8. This finding indicates inhomogeneous horizontal distributions of the pollutants. In order to show the phenomenon more clearly, we plotted 3D distribution of aircraft data on 21 May in the following figures. In the figure, the colors indicate aerosol extinctions or VMRs of $NO_2$ and $SO_2$. The black dots on the surface represent the location of the MAX-DOAS instrument, and the arrows point to the direction of the MAX-DOAS telescope. The figures clearly indicate strong horizontal gradients of the pollutants. The new figures given below are not included in the manuscript because its information can be well shown with the variability of original aircraft data in Fig. 8. Since in the pointing direction of the MAX-DOAS telescope only a few aircraft measurements are available, we didn't do the comparisons only for these data. Another important aspect is that the aircraft results are from in-situ measurements, whereas MAX-DOAS measurements represent averages of the pollutants along an effective light path of ~ 5 to 10 km. Therefore the different air masses measured by the two techniques can be seen as one important reason for the differences of the results. The effect of different probed air masses was not clearly pointed out in the previous manuscript. Therefore we added the following sentences in the revised manuscript:
"In addition, aircraft results represent in-situ measurements along the spiral route, whereas MAX-DOAS results represent averages of pollutants over an effective light path of ~5 to 10 km. The different air masses measured by the two techniques can be seen as one important reason for the observed differences of the measured results."

[Figure]

3) - In addition to co-author's research group, other aircraft measurement work in the NCP region should be credited, e,g. Ma et al. (2012) and Zhang et al. (2014); so did the MAX-DOAS measurement, e.g., Jin et al. (2016).

Author reply: Thanks for reminding these references! We cited them in appropriate positions in the introduction section of the revised manuscript.

Technical corrections:

1) Page 2, Line 32. What does "East-Aire" mean ?
   Author reply: It is the abbreviation of "East Asian Study of Tropospheric Aerosols: an International Experiment". The full name is given in the revised manuscript.

2) Page 4, Line 23: What is the terrain height of the station?
   Author reply: The terrain height of the station is ~200 m asl. The information is given in the revised manuscript.

3) Page 5, Line 15-20: The direction for the measurement should be mentioned.
   Author reply: The telescope was pointed to an azimuth angle of 25° northeast. The information is given in section 2.2.1 in the revised manuscript.

4) Page 7, Line 12. Please check the punctuation here as well as elsewhere in the manuscript.
   Author reply: the punctuations are checked.

5) Page 8, Line 7-8. There are two references of Zhang et al., 2018. Please distinguish them when citing.
   Author reply: One of the tow references was wrongly cited. Therefore we deleted this one in the revised manuscript.

6) Page 9, Line 15-17. Please delete the repeating word of "be".
   Author reply: It is corrected in the revised manuscript.

7) Page 12, Line 25-26. The agreement of the aerosol profiles from MAX-DOAS and lidar above 500m is not obvious, especially on 16 May, 2016. It is better to alter the color bar to show this point more clearly.
   Author reply: Since aerosol extinction at high altitudes is much lower than those at ~500m, we think it is not necessary to highlight these differences. We prefer to use the current color bar, which can balance the requirement to show structures of high concentrations and low concentrations at different altitudes.

8) Page 12, Line 30. The content should move to the section 4.1.
   Author reply: We added the following clarification in the section 4.1: "In addition, the sun-photometer measured the air masses in the direction of the sun. The different air mass measured by the two techniques can contribute to the differences of the AOD results."
   We also clarified the same statements for the other instruments in section 4.1 as follows: "Note that the visibilitymeter and the in-situ measurements of NO2, SO2 and HCHO represent air masses close to the instruments, whereas the MAX-DOAS measurements represent averages of pollutants along the effective horizontal light path of ~5 to 10 km in the vertical grid from the surface up to 0.2 km. Therefore different probed air masses can be seen as one important reason for the differences of results."

9) Page 13, Line 20-22. This paragraph seems to be redundant.
   Author reply: We prefer to keep the paragraph to give readers an impression on the contents of section 5 before they go to the details.

10) Page 14, Line 4. The language expression needs to be improved.
    Author reply: The sentence was modified in the revised manuscript as follows: The mountain-plain topography causes a daily cycle with downslope (northeast winds) and upslope (southeast winds) winds.

11) Page 16, Line 6. Please note the subscript.
    Author reply: They are corrected in the revised manuscript.

12) Page 16: Sect. 5.2.2. It is known that the MAX-DOAS measurements are per- formed during the daytime. However, the sorting here is mainly based on the nighttime trajectories. In addition, there are large differences between nighttime and daytime in Fig.S5, especially for the southerly trajectories.
Author reply: We clarified the reason why we use the nighttime trajectories to separate the results in section 5.1 as follows "Therefore we can expect that the gas pollutants, e.g. NOx, SO2 and HONO, can be transported to a farther distance during nighttime than daytime. Thus nighttime regional transport of pollutants from the Wuan area could significantly pollute the entire measurement area. " and in section 5.2.2 "Considering that the life times of the observed trace gases are typically longer during night time than day time (because of lower OH radical concentrations), the measurement data are sorted mainly based on the nighttime trajectories". Note that generally the difference between the nighttime and daytime trajectories in Fig. S5 is not big. But the difference is also the reason why we only use night time trajectories to sort the data.

13) Page 17, Line 34. Please change "Fig.12i" to "Fig.12I".
Author reply: It is changed in the revised manuscript.

14) Page 22, Line 23, 26. Please check the names of the station "Wuxi" and software "WINDOAS".
Author reply: They are all corrected in the "Acknowledgements" of the revised manuscript.

15) Figure 6: Please clarify the temporal resolution of the data used in Fig.6.
Author reply: we added the information for the figure in the revised manuscript as follows:
"All independent data are averaged over the individual time intervals of the MAX-DOAS measurements."

16) Table 2: "outliers"?
Author reply: We used the filters given in the table to get rid of "outliers". We do not apply any other filters to delete specific "outliers".

17) References
Jin, J., Ma, J., Lin, W., Zhao, H., Shaiganfar, R., Beirle, S., and Wagner, T.: MAX-DOAS measurements and satellite validation of tropospheric $NO_2$ and $SO_2$ vertical column densities at a rural site of North China, Atmospheric Environment, 133, 12-25, http://dx.doi.org/10.1016/j.atmosenv.2016.03.031, 2016.
Ma, J. Z., Wang, W., Chen, Y., Liu, H. J., Yan, P., Ding, G. A., Wang, M. L., Sun, J., and Lelieveld, J.: The IPAC-NC field campaign: a pollution and oxidization pool in the lower atmosphere over Huabei, China, Atmos. Chem. Phys., 12, 3883-3908, 10.5194/acp- 12-3883-2012, 2012.
Zhang, W., Zhu, T., Yang, W., Bai, Z., Sun, Y. L., Xu, Y., Yin, B., and Zhao, X.: Airborne measurements of gas and particle pollutants during CAREBeijing-2008, Atmos. Chem. Phys., 14, 301-316, 10.5194/acp-14-301-2014, 2014.

Author reply: Thanks for reminding these references. We cited them at appropriate positions in the introduction section of the revised manuscript.